



# A High-order Staggered Finite-Element Vertical Discretization for Non-Hydrostatic Atmospheric Models

## J. E. Guerra and P. A. Ullrich

Jorge Guerra, Department of Land, Air and Water Resources, University of California, Davis, One Shields Ave., Davis, CA 95616. Email: jeguerra@ucdavis.edu

Correspondence to: Jorge E. Guerra
(jeguerra@ucdavis.edu)



## Abstract

Atmospheric modeling systems require economical methods to solve the non-hydrostatic Euler
equations. Two major differences between hydrostatic models and a full non-hydrostatic de-
scription lies in the vertical velocity tendency and numerical stiffness associated with sound
waves. In this work we introduce a new arbitrary-order vertical discretization entitled the Stag-
gered Nodal Finite Element Method (SNFEM). Our method uses a generalized discrete deriva-
tive that consistently combines the Discontinuous Galerkin and Spectral Element methods on
a staggered grid. Our combined method leverages the accurate wave propagation and conser-
vation properties of spectral elements with staggered methods that eliminate stationary $(2\Delta x)$
modes. Furthermore, high-order accuracy also eliminates the need for a reference state to main-
tain hydrostatic balance. In this work we demonstrate the use of high vertical order as a means
of improving simulation quality at relatively coarse resolution. We choose a test case suite that
spans the range of atmospheric flows from predominantly hydrostatic to nonlinear in the Large
Eddy regime. Our results show that there is a distinct benefit in using the high-order vertical
coordinate at low resolutions with the same robust properties of the low-order alternative.

## 1 Introduction

The accurate representation of vertical wave motion is essential for models of the atmosphere.
The vertical coordinate for the non-hydrostatic fluid equations has traditionally been discretized
in the Eulerian frame via a second-order Charney-Phillips (Charney and Phillips, 1953) or
Lorenz grid (Arakawa and Moorthi, 1988), or via Lagrangian layers, such as in Lin (2004).
However, little work has been undertaken to develop high-order vertical discretizations due to a
number of outstanding issues. First, higher-order generalizations must somehow incorporate the
no-flux boundary conditions at the model bottom and top without loss of accuracy, especially
near the surface where accurate treatment of dynamics is paramount. Second, as observed by
Thuburn and Woollings (2005), Thuburn (2006) and Toy and Randall (2007) the choice of verti-
cal coordinate (whether height-based, mass-based or entropy-based) implies an optimal vertical





staggering of prognostic variables for maintaining correct behavior for wave motions relevant to the atmosphere. Third, unstaggered discretizations (that is, discretizations where all prognostic variables are stored on model levels) possess stationary computational modes which represent gross errors in the dispersion properties of the solution (Melvin et al., 2012; Ullrich, 2014b).

5 As in the horizontal, unstaggered FEM leads to waves with zero phase speed in the limit as the wavelength tends to $2\Delta x$, where $\Delta x$ is the average grid spacing between degrees of freedom. However, unlike the horizontal, these wave modes can be dramatically enhanced by an implicit treatment of the vertical at high Courant number.

This paper describes a new discretization for the vertical that combines the accuracy of fi-

10 nite element methods with the desirable wave propagation properties of staggered methods. This approach is referred to as the Staggered Nodal Finite Element Method (SNFEM). Notably, this formulation is sufficiently general to be compatible with essentially any form of the fluid equations. The SNFEM discretization can be easily composed in differential form using interpolation and differentiation operators built in accordance with the Discontinuous Galerkin

15 and Spectral Element discretizations that arise from the Flux Reconstruction method of Huynh (2007) (see Table 1). The use of SNFEM is natural for vertical discretizations, as no-flux conditions are easily imposed on top and bottom boundaries in the general finite element framework (Zienkiewicz et al., 2005). Also, the structure of individual elements (namely, the fact that Gauss and Gauss-Lobatto-Legendre nodes are optimally clustered near element edges) lends it-

20 self readily to improved resolution in the atmospheric boundary layer. Further, SNFEM inherits the mimetic properties of the spectral element method so the vertical operator will automatically conserve both mass and discrete linear energy.

To assess the performance of SNFEM, this discretization has been implemented in the spectral element Tempest model (Ullrich, 2014a) and run through a suite of mesoscale test cases.

25 The test cases are as follows: A rising thermal convective bubble (Giraldo and Restelli, 2008), the density current of Straka et al. (1993), uniform flow over the mountain of Schär et al. (2002), and the baroclinic instability in a 3D Cartesian channel of Ullrich et al. (2015). While not exhaustive, this validation suite is intended to show the treatment of waves, non-linear vertical





transport, and near boundary dynamics corresponding to a high-order vertical coordinate with and without the influence of topography. Therefore, the objectives of this paper are as follows:

1. To introduce our approach for the construction of a generalized, staggered, variable order-of-accuracy, finite element vertical discretization. We emphasize discretization of the non-conservative differential form of the Navier-Stokes equations (in vector invariant or so-called Clark form), which is independent of coordinate system.

2. To validate the implementation of this discretization within the Tempest framework using a selection of test cases in Cartesian geometry through a range of horizontal scales from 1 to 1000 km.

3. To determine the qualitative and quantitative effect of vertical order of accuracy on solutions by conducting validation experiments at coarse resolutions relative to finer reference solutions. We consider the effects of Lorenz (LOR) and Charney-Phillips (CPH) staggering both in the interior flow and at the lower boundary.

4. To determine whether a high-order vertical discretization greatly improves the simulation quality, and consequently to recommend whether there is an optimal order-of-accuracy that provides the best tradeoff between accuracy and computational cost.

We will show that a high-order vertical discretization at coarse resolution more accurately approximates the reference solution relative to the low vertical order alternative. Since the interpolation and derivative operators in the finite element approach are easily expressed as linear matrix operators, there is minimal cost in adjusting the order-of-accuracy. We will present control experiments 4 where only the resolution and vertical order-of-accuracy vary. We leave the rigorous analysis of staggered wave modes and discrete energy conservation using the interpolation/differentiation operators for a subsequent work.

The remainder of this manuscript is as follows: Section 2 describes the non-hydrostatic equations of fluid motion on an arbitrary coordinate frame. Section 3 describes the discrete form of these equations, including the spectral element horizontal discretization, the operators used



by the SNFEM vertical discretization and the time-stepping scheme employed. In section 4, we describe the test case suite and discuss the corresponding model results. The summary and conclusions follow in section 5.

## 2   The non-hydrostatic equations of fluid motion

In an arbitrary coordinate frame $(\alpha, \beta, \xi)$, the vector velocity can be written as

$$\mathbf{u} = u^{\alpha}\mathbf{g}_{\alpha} + u^{\beta}\mathbf{g}_{\beta} + u^{\xi}\mathbf{g}_{\xi}, \tag{1}$$

where $\boldsymbol{g}_i$ $(i \in \{\alpha, \beta, \xi\})$ are the local coordinate basis vectors and $u^i$ are the contravariant velocity components. The associated covariant components are

$$u_{\alpha} = \mathbf{u} \cdot \mathbf{g}_{\alpha}, \qquad u_{\beta} = \mathbf{u} \cdot \mathbf{g}_{\beta}, \qquad u_{\xi} = \mathbf{u} \cdot \mathbf{g}_{\xi}. \tag{2}$$

Covariant components can be obtained in terms of contravariant components via contraction with the covariant metric $g_{ij} = \mathbf{g}_i \cdot \mathbf{g}_j$,

$$u_i = g_{i\alpha}u^{\alpha} + g_{i\beta}u^{\beta} + g_{i\xi}u^{\xi}. \tag{3}$$

The reverse operation uses the contravariant metric $g^{ij}$, defined as the matrix inverse of the covariant metric. Contraction of the covariant components with the contravariant metric returns the contravariant vector components,

$$u^i = g^{i\alpha}u_{\alpha} + g^{i\beta}u_{\beta} + g^{i\xi}u_{\xi}. \tag{4}$$

The volume element $J$ is computed in terms of the covariant metric as

$$J = \sqrt{\det g_{ij}}. \tag{5}$$

Using covariant horizontal velocity components, vertical velocity, potential temperature $\theta$ and dry air density $\rho$ as prognostic variables, the Euler equations with shallow-atmosphere



approximation can be written an arbitrary coordinate frame as

$$\frac{\partial u_\alpha}{\partial t} = -\frac{\partial}{\partial \alpha}(K + \Phi) - \theta\frac{\partial \Pi}{\partial \alpha} + (\boldsymbol{\eta} \times \mathbf{u})_\alpha, \tag{6}$$

$$\frac{\partial u_\beta}{\partial t} = -\frac{\partial}{\partial \beta}(K + \Phi) - \theta\frac{\partial \Pi}{\partial \beta} + (\boldsymbol{\eta} \times \mathbf{u})_\beta, \tag{7}$$

$$\left(\frac{\partial r}{\partial \xi}\right)\frac{\partial w}{\partial t} = -\frac{\partial}{\partial \xi}(K + \Phi) - \theta\frac{\partial \Pi}{\partial \xi} + (\boldsymbol{\eta} \times \mathbf{u})_\xi \tag{8}$$

$$\frac{\partial \theta}{\partial t} = -u^\alpha\frac{\partial \theta}{\partial \alpha} - u^\beta\frac{\partial \theta}{\partial \beta} - u^\xi\frac{\partial \theta}{\partial \xi}, \tag{9}$$

$$\frac{\partial \rho}{\partial t} = -\frac{1}{J}\frac{\partial}{\partial \alpha}(J\rho u^\alpha) - \frac{1}{J}\frac{\partial}{\partial \beta}(J\rho u^\beta) - \frac{1}{J}\frac{\partial}{\partial \xi}(J\rho u^\xi). \tag{10}$$

The vertical velocity $w$ is closely related to $u_\xi$ via

$$w = |\mathbf{g}_\xi|^{-1}u_\xi, \tag{11}$$

The specific Kinetic energy is

$$K = \frac{1}{2}\left(u_\alpha u^\alpha + u_\beta u^\beta + u_\xi u^\xi\right), \tag{12}$$

while the geopotential function given the gravitational acceleration $g_c$ is

$$\Phi = g_c r(\xi), \tag{13}$$

and the Exner pressure is

$$\Pi = c_p\left(\frac{p_0}{p}\right)^{R_d/c_p} = c_p\left(\frac{R_d\rho\theta}{p_0}\right)^{R_d/c_v}. \tag{14}$$

Here $p_0$ denotes the constant reference pressure, $R_d$ is the ideal gas constant and $c_v$ and $c_p$ refer to the specific heat capacity at constant volume and pressure, respectively. The absolute vorticity vector is given by

$$\boldsymbol{\eta} = \boldsymbol{\zeta} + \boldsymbol{\omega}, \tag{15}$$





where the relative vorticity vector is

$$\boldsymbol{\zeta} = \frac{1}{J}\left[\left(\frac{\partial u_\xi}{\partial \beta} - \frac{\partial u_\beta}{\partial \xi}\right)\mathbf{g}_\alpha + \left(\frac{\partial u_\alpha}{\partial \xi} - \frac{\partial u_\xi}{\partial \alpha}\right)\mathbf{g}_\beta + \left(\frac{\partial u_\beta}{\partial \alpha} - \frac{\partial u_\alpha}{\partial \beta}\right)\mathbf{g}_\xi\right], \tag{16}$$

and, under the shallow-atmosphere approximation, the planetary vorticity vector is

$$\boldsymbol{\omega} = f(\partial r/\partial \xi)^{-1}\mathbf{g}_\xi. \tag{17}$$

Consequently, the rotational terms in the equation of motion take the form

$$(\eta \times \mathbf{u})_\alpha = J\left[u^\beta(\omega^\xi + \zeta^\xi) - u^\xi\zeta^\beta\right], \tag{18}$$

$$(\eta \times \mathbf{u})_\beta = J\left[u^\xi\zeta^\alpha - u^\alpha(\omega^\xi + \zeta^\xi)\right], \tag{19}$$

$$(\eta \times \mathbf{u})_\xi = J\left[u^\alpha\zeta^\beta - u^\beta\zeta^\alpha\right]. \tag{20}$$

Note that this formulation does not specify a coordinate system. Consequently, these equations can be used for either Cartesian or spherical geometry. To account for topography, terrain-following $\sigma$-coordinates are imposed by defining the radius $r = r(\alpha, \beta, \xi)$ so that $r(\alpha, \beta, 0)$ is coincident with the surface. For example, Gal-Chen and Somerville (1975) coordinates arise from the choice

$$r(\alpha, \beta, \xi) = \xi\left[z_{top} - z_s(\alpha, \beta)\right] + r_e + z_s(\alpha, \beta), \tag{21}$$

where $r_e$ is the radius of the Earth, $z_{top}$ denotes the model height and $z_s(\alpha, \beta)$ denotes the surface elevation.

# 3 Discretization

## 3.1 Horizontal Discretization

Spherical shells are discretized using an equiangular cubed-sphere grid (Sadourny, 1972; Ronchi et al., 1996), which consists of six Cartesian patches arranged along the faces of a cube which



is then inflated onto a spherical shell. More information on this choice of grid can be found in Ullrich (2014a). On the equiangular cubed-sphere grid, coordinates are given as $(\alpha, \beta, p)$, with central angles $\alpha, \beta \in [-\frac{\pi}{4}, \frac{\pi}{4}]$ and panel index $p \in \{1, 2, 3, 4, 5, 6\}$. By convention, we choose panels 1–4 to be along the equator and panels 5 and 6 to be centered on the northern and southern pole, respectively. With uniform grid spacing, each panel consists of a square array of $n_e \times n_e$ finite elements.

Spherical coordinates $(\lambda, \phi)$ for longitude $\lambda \in [0, 2\pi]$ and latitude $\phi \in [-\pi/2, \pi/2]$ will also be used for plotting and specification of tests. Coordinate transforms between spherical and equiangular coordinates can be found in Ullrich and Jablonowski (2012) Appendix A.

The horizontal discretization of (6)-(10) follows the continuous element formulation of Ullrich (2014a), which is analogous to earlier efforts with spectral elements (Giraldo and Rosmond, 2004; Taylor and Fournier, 2010; Dennis et al., 2011; Giraldo et al., 2013).

## 3.2 Vertical Discretization

Each vertical column consists of $n_{ve}$ nodal finite elements, indexed $a \in \{0, \ldots, n_{ve}-1\}$. Throughout this manuscript, all vertical indices are assumed to increase with altitude. Within each element, levels are placed at the $n_{vp}$ Gaussian quadrature nodes and interfaces at $n_{vp} + 1$ Gauss-Lobatto quadrature nodes, leading to a staggering of levels and interfaces. With vertical coordinate $\xi$, the location of model levels denoted $\xi_{a,k}$ with $k \in \{0, \ldots, n_{vp} - 1\}$ and model interfaces denoted $\tilde{\xi}_{a,k}$ with $k \in \{0, \ldots, n_{vp}\}$. Each finite element is then bounded within the interval $[\tilde{\xi}_{a,0}, \tilde{\xi}_{a,n_{vp}}]$ with two associated sets of basis functions – one over model levels, denoted by the set $\phi_a = \{\phi_{a,j} | j = 0, \ldots, n_{vp} - 1\}$ that includes characteristic polynomials of degree $n_{vp} - 1$, and one over model interfaces, denoted by the set $\tilde{\phi}_a = \{\tilde{\phi}_{a,j} | j = 0, \ldots, n_{vp} - 1\}$ that includes characteristic polynomials of degree $n_{vp}$. A depiction of the vertical staggering associated with levels and interfaces is given in Fig. 1, along with basis functions in each case. A scalar field $q(\xi, t)$ can then be written approximately, either as a linear combination of basis functions on



levels,

$$q(\xi,t) \approx \sum_{a=0}^{n_{ve}-1} \sum_{j=0}^{n_{vp}-1} q_{a,j}(t)\phi_{a,j}(\xi), \tag{22}$$

or on interfaces,

$$q(\xi,t) \approx \sum_{a=0}^{n_{ve}-1} \sum_{j=0}^{n_{vp}} \tilde{q}_{a,j}(t)\tilde{\phi}_{a,j}(\xi). \tag{23}$$

For the remainder of this manuscript we will use script $n$ to denote variables stored on model levels and script $i$ to denote variables stored on interfaces.

### 3.2.1 Interpolation Operators

Note that (22) and (23) are not equivalent discretizations since (22) cannot represent polynomials of degree $n_{vp}$ and (23) cannot represent fields that are discontinuous at element interfaces. Nonetheless, we can define interpolation operators between these fields via $\mathcal{I}_i^n$, representing interpolation from levels to interfaces, and $\mathcal{I}_n^i$, representing interpolation from interfaces to nodes. First, interpolation from interfaces to levels is defined as

$$(\mathcal{I}_n^i \tilde{\mathbf{q}})_{a,k} = \sum_{j=0}^{n_{vp}} \tilde{q}_{a,j}(t)\tilde{\phi}_{a,j}(\xi_{a,k}). \tag{24}$$

To define the interpolant from levels to interfaces, a two-step procedure is employed. Since basis functions on levels are discontinuous, we define the left and right interpolants at element boundaries as

$$(\mathcal{I}_L^n \mathbf{q})_{a,0} = \sum_{j=0}^{n_{vp}-1} q_{a,j}\phi_{a,j}(\tilde{\xi}_{a,0}), \qquad (\mathcal{I}_R^n \mathbf{q})_{a,n_{vp}-1} = \sum_{j=0}^{n_{vp}-1} q_{a,j}\phi_{a,j}(\tilde{\xi}_{a,n_{vp}-1}) \tag{25}$$



and then define the total interpolant as

$$
(\mathcal{I}_i^n \mathbf{q})_{a,k} =
\begin{cases}
\displaystyle\sum_{j=0}^{n_{vp}-1} q_{a,j}\phi_{a,j}(\tilde{\tilde{\xi}}_{a,k}) & \text{if } 0 < k < n_l, \\[12pt]
\frac{1}{2}(\mathcal{I}_R^n \mathbf{q})_{a-1,n_{vp}-1} + \frac{1}{2}(\mathcal{I}_L^n \mathbf{q})_{a,0} & \text{if } k = 0, \\[12pt]
\frac{1}{2}(\mathcal{I}_R^n \mathbf{q})_{a,n_{vp}-1} + \frac{1}{2}(\mathcal{I}_L^n \mathbf{q})_{a+1,0} & \text{if } k = n_l.
\end{cases}
\tag{26}
$$

These interpolation operators can also be obtained from equivalence via the variational (weak) form. At model interfaces, the accuracy of (26) degrades for unequally spaced finite elements. For the case of stacked finite elements with unequal thickness $\Delta\xi_a = \tilde{\xi}_{a,n_{vp}} - \tilde{\xi}_{a,0}$, a more accurate formula can be obtained from

$$
(\mathcal{I}_i^n \mathbf{q})_{a,0} = \frac{\Delta\xi_a^{n_{vp}}(\mathcal{I}_R^n \mathbf{q})_{a-1,n_{vp}-1} + \Delta\xi_{a-1}^{n_{vp}}(\mathcal{I}_L^n \mathbf{q})_{a,0}}{\Delta\xi_a^{n_{vp}} + \Delta\xi_{a-1}^{n_{vp}}},
\tag{27}
$$

which arises on noting that the one-sided interpolant has error $O(\Delta\xi_a^{n_{vp}})$.

### 3.2.2 Differentiation Operators

Differentiation is required for all combinations of model levels and interfaces: $\mathcal{D}_i^i$ represents differentiation from interfaces to interfaces, $\mathcal{D}_n^i$ represents differentiation from interfaces to levels, $\mathcal{D}_n^n$ denotes differentiation from levels to levels and $\mathcal{D}_i^n$ denotes differentiation from levels to interfaces. A depiction of the behavior of these derivative operators is shown in Fig. 2.

Differentiation from interfaces to levels is obtained by simply differentiating (24),

$$
(\mathcal{D}_n^i \mathbf{q})_{a,k} = \sum_{j=0}^{n_l} \tilde{q}_j \frac{\partial \tilde{\phi}_j}{\partial \xi}(\xi_{a,k})
\tag{28}
$$

Differentiation from levels to levels is computed via the composed operator

$$
\mathcal{D}_n^n \mathbf{q} = \mathcal{D}_n^i \mathcal{I}_i^n \mathbf{q},
\tag{29}
$$





where boundary conditions are enforced after application of the interpolation operator.

Differentiation from interfaces to interfaces requires averaging the one-sided derivatives at element interfaces, but is otherwise simply the derivative of (24) on the element interior,

$$
(\mathcal{D}_i^i \mathbf{q})_{a,k} =
\begin{cases}
\dfrac{1}{2}\left( \displaystyle\sum_{j=0}^{n_{vp}} \tilde{q}_{a,j} \dfrac{\partial \tilde{\phi}_{a,j}}{\partial \xi}(\tilde{\xi}_{a,k}) + \sum_{j=0}^{n_{vp}} \tilde{q}_{a-1,j} \dfrac{\partial \tilde{\phi}_{a-1,j}}{\partial \xi}(\tilde{\xi}_{a,k}) \right) & \text{if } k = 0, \\[3ex]
\displaystyle\sum_{j=0}^{n_{vp}} \tilde{q}_{a,j} \dfrac{\partial \tilde{\phi}_{a,j}}{\partial \xi}(\tilde{\xi}_{a,k}) & \text{if } 0 < k < n_l, \\[3ex]
\dfrac{1}{2}\left( \displaystyle\sum_{j=0}^{n_{vp}} \tilde{q}_{a,j} \dfrac{\partial \tilde{\phi}_{a,j}}{\partial \xi}(\tilde{\xi}_{a,k}) + \sum_{j=0}^{n_{vp}} \tilde{q}_{a+1,j} \dfrac{\partial \tilde{\phi}_{a+1,j}}{\partial \xi}(\tilde{\xi}_{a,k}) \right) & \text{if } k = n_l.
\end{cases}
\tag{30}
$$

Differentiation from levels to interfaces ($\mathcal{D}_i^n$) should not be defined via the composition $\mathcal{D}_i^i \mathcal{I}_i^n$ since this procedure would introduce a non-zero null space that can trigger an unphysical computational mode in the discrete equations. Instead we define $\mathcal{D}_i^n$ using the robust differentiation technique discussed in Ullrich (2014a), based on the flux reconstruction methods of Huynh (2007). This strategy leads to the discrete operator

$$
(\mathcal{D}_i^n \mathbf{q})_{a,k} = (\hat{\mathcal{D}}_i^n \mathbf{q})_{a,k} + \frac{1}{2}\frac{dg_R}{d\xi}(\tilde{\xi}_{a,k}) \left[ (\mathcal{I}_L^n \mathbf{q})_{a+1,k} - (\mathcal{I}_R^n \mathbf{q})_{a,k} \right]
$$

$$
+ \frac{1}{2}\frac{dg_L}{d\xi}(\tilde{\xi}_{a,k}) \left[ (\mathcal{I}_R^n \mathbf{q})_{a-1,k} - (\mathcal{I}_L^n \mathbf{q})_{a,k} \right],
\tag{31}
$$





where

$$
(\hat{\mathcal{D}}_i^n \mathbf{q})_{a,k} =
\begin{cases}
\dfrac{1}{2}\left( \displaystyle\sum_{j=0}^{n_{vp}-1} q_{a,j}\dfrac{\partial \phi_{a,j}}{\partial \xi}(\tilde{\xi}_{a,k}) + \sum_{j=0}^{n_{vp}-1} q_{a-1,j}\dfrac{\partial \phi_{a-1,j}}{\partial \xi}(\tilde{\xi}_{a,k}) \right) & \text{if } k = 0, \\[2em]
\displaystyle\sum_{j=0}^{n_{vp}-1} q_{a,j}\dfrac{\partial \phi_{a,j}}{\partial \xi}(\tilde{\xi}_{a,k}) & \text{if } 0 < k < n_l, \\[2em]
\dfrac{1}{2}\left( \displaystyle\sum_{j=0}^{n_{vp}-1} q_{a,j}\dfrac{\partial \phi_{a,j}}{\partial \xi}(\tilde{\xi}_{a,k}) + \sum_{j=0}^{n_{vp}-1} q_{a+1,j}\dfrac{\partial \phi_{a+1,j}}{\partial \xi}(\tilde{\xi}_{a,k}) \right) & \text{if } k = n_l,
\end{cases}
\tag{32}
$$

and $g_L$ and $g_R$ are the local *flux correction functions*, which are chosen to satisfy

$$
g_L(\xi_{a,0}) = 1, \qquad g_L(\xi_{a,n_{vp}-1}) = 0, \qquad g_R(\xi_{a,0}) = 0, \qquad g_R(\xi_{a,n_{vp}-1}) = 1, \tag{33}
$$

and otherwise approximate zero throughout $[\xi_{a,0}, \xi_{a,n_{vp}-1}]$.

There is some flexibility in the discretization that depends on the specific choice of flux correction functions. Huynh (2007) gives a family of flux correction functions on the interval $[-1,1]$ denoted by $g_k$ for $k = 1, 2, \ldots$. In particular, we are interested in $g_1$ (the Radau polynomials) and $g_2$, which have the special property that $dg_2/dx = 0$ at all Gauss-Lobatto points. Although either choice of flux correction function leads to a valid discretization for $n_{vp} > 1$, when $n_{vp} = 1$ a consistent differential operator is recovered only with $g_2$. Hence, for the remainder of this text we will adopt the flux correction function $g_2$. For this choice, the flux correction function satisfies

$$
\frac{\partial g_2}{\partial x} = \frac{(n_{vp}+1)\left[P_{n_{vp}+1}(x) - xP_{n_{vp}}(x)\right]}{2(x-1)}, . \tag{34}
$$

where $P_N(x)$ is the Legendre polynomial of order $N$. In the limit as $x$ approaches the boundaries of the reference element, a simplified expression emerges:

$$
\frac{\partial g_2}{\partial x}(x \to +1) = n_{vp}(n_{vp}+1). \tag{35}
$$





On the interval $[\tilde{\xi}_{j,0}, \tilde{\xi}_{j,n_{vp}-1}]$ we have

$$\frac{\partial g_R}{\partial \xi}(\xi) = \frac{1}{\Delta \xi_a} \frac{\partial g_2}{\partial x}\left[\frac{2(\xi - \xi_{j,0})}{\Delta \xi_a} - 1\right], \quad \frac{\partial g_L}{\partial \xi}(\xi) = -\frac{1}{\Delta \xi_a} \frac{\partial g_2}{\partial x}\left[\frac{2(\xi_{j,n_{vp}-1} - \xi)}{\Delta \xi_a} - 1\right]. \quad (36)$$

### 3.2.3 Second Derivative Operators in the Vertical

The second derivative operators are used in viscosity and hyperviscosity calculations. They are obtained as approximations to the equation

$$\mathcal{L}(\nu)\mathbf{q} \approx \nu \frac{\partial^2 q}{\partial \xi^2}, \quad (37)$$

subject to Neumann (no-flux) boundary condition

$$\frac{\partial q}{\partial \xi} = 0 \quad \text{at } \xi = 0 \text{ and } \xi = 1. \quad (38)$$

For the viscous operator from interfaces to interfaces, the discretization is obtained from the variational (weak) formulation. Specifically, from (37) and integration by parts,

$$\int_0^1 (\mathcal{L}_i^i \mathbf{q})_{b,n} \tilde{\phi}_{a,k} d\xi = \left. \frac{\partial q}{\partial \xi} \tilde{\phi}_{a,k}\right|_0^1 - \int_0^1 \frac{\partial q}{\partial \xi} \frac{\partial \tilde{\phi}_{a,k}}{\partial \xi} d\xi. \quad (39)$$

Then using (23), (38) and the assumption of orthogonality of basis functions $\tilde{\phi}$ under quadrature,

$$(\mathcal{L}_i^i \mathbf{q})_{a,k} = -\frac{1}{\int_0^1 \tilde{\phi}_{a,k}^2 d\xi} \sum_{b=0}^{n_{ve}-1} \sum_{n=0}^{n_{vp}} \tilde{q}_{b,n} \int_0^1 \frac{\partial \tilde{\phi}_{a,k}}{\partial \xi} \frac{\partial \tilde{\phi}_{b,n}}{\partial \xi} d\xi. \quad (40)$$

For model interfaces on Gauss-Lobatto nodes, the integral is discretized via Gauss-Lobatto quadrature.





The viscous operator from levels to levels is derived in a similar manner, although the non-differentiability of $q$ at interfaces in the discontinuous basis means that we must rely on differentiation via (31). Consequently, the weak form

$$\int_{\tilde{\xi}_{a,0}}^{\tilde{\xi}_{a,v_{np}}} (\mathcal{L}_i^i \mathbf{q})_{b,n} \phi_{a,k} d\xi = \frac{\partial q}{\partial \xi} \phi_{a,k} \Big|_{\tilde{\xi}_{a,0}}^{\tilde{\xi}_{a,v_{np}}} - \int_{\tilde{\xi}_{a,0}}^{\tilde{\xi}_{a,v_{np}}} \frac{\partial q}{\partial \xi} \frac{\partial \phi_{a,k}}{\partial \xi} d\xi. \tag{41}$$

then leads to discrete operator

$$(\mathcal{L}_n^n \mathbf{q})_{a,k} = \frac{1}{\int_{\tilde{\xi}_{a,0}}^{\tilde{\xi}_{a,n_{vp}}} \phi_{a,k}^2 d\xi} \left[ (\hat{\mathcal{L}}_n^n \mathbf{q})_{a,k} + (\mathcal{D}_i^n \mathbf{q})_{a,v_{np}} \phi(\tilde{\xi}_{a,v_{np}}) - (\mathcal{D}_i^n \mathbf{q})_{a,0} \phi(\tilde{\xi}_{a,0}) \right], \tag{42}$$

where

$$(\hat{\mathcal{L}}_n^n \mathbf{q})_{a,k} = - \sum_{b=0}^{n_{ve}-1} \sum_{n=0}^{n_{vp}-1} q_{b,n} \int_{\tilde{\xi}_{a,0}}^{\tilde{\xi}_{a,n_{vp}}} \frac{\partial \phi_{a,k}}{\partial \xi} \frac{\partial \phi_{b,n}}{\partial \xi} d\xi. \tag{43}$$

For model levels on Gauss nodes, the integral is discretized directly via Gaussian quadrature. Note that the boundary condition implies that we must impose

$$(\mathcal{D}_i^n \mathbf{q})_{0,0} = 0 \quad \text{and} \quad (\mathcal{D}_i^n \mathbf{q})_{v_{ne}-1,v_{np}} = 0. \tag{44}$$

### 3.2.4 Flow-dependent vertical hyperviscosity

The basic spectral element method is an energy conservative scheme (Taylor and Fournier, 2010) that allows for the accumulation of energy at the shortest wavelengths. Following Ullrich (2014a) and Dennis et al. (2011), we impose explicit dissipation in the horizontal using a constant coefficient hyperviscosity. In the vertical, a constant coefficient hyperviscosity would have





a rapid and adverse affect on hydrostatic balance in the absence of a hydrostatic reference state (Giraldo and Restelli, 2008). Consequently, in this paper we apply a localized hyperviscosity in the vertical column that is weighted by the contravariant vertical flow velocity $u^\xi$,

$$\frac{\partial q}{\partial t} = \cdots + \nu_z |u^\xi| \frac{\partial^{2k} q}{\partial \xi^{2k}}, \tag{45}$$

where $q \in \{u_\alpha, u_\beta, w, \theta, \rho\}$ and $k$ is a positive integer. The motivation for using $u^\xi$ stems from the observation that advective transport in the vertical occurs with speed $u^\xi$, and so this would be the corresponding wave speed that would enter into, for example, the Rusanov Riemann solver in the context of discontinuous Galerkin or finite volume methods. In this sense, the flow-dependent hyperviscosity is a generalization of advective up-winding if applied simultaneously with the vertical advective operator. The Riemann solver interpretation also yields an appropriate estimate for the value of $\nu_z$,

$$
\begin{aligned}
k = 2: & \quad \nu_z = (1/2)(\overline{\Delta\xi})^{-1}, \\
k = 4: & \quad \nu_z = -(1/12)(\overline{\Delta\xi})^{-3}, \\
k = 6: & \quad \nu_z = (1/60)(\overline{\Delta\xi})^{-5},
\end{aligned}
\tag{46}
$$

where $\overline{\Delta\xi} = 1/(an_{vp})$ is the average spacing between nodes in the vertical direction.

### 3.2.5 The Staggered Nodal Finite Element Method (SNFEM)

The interpolation and differentiation operators given in the previous sections provide a framework for constructing staggered vertical grids in the context of the nonlinear system (6)-(10). Furthermore, the SNFEM allows for discretizations of arbitrary order-of-accuracy via adjustments in $n_{vp}$. For the present work, we investigate unstaggered (on interfaces), Lorenz (LOR) ($u, v, \rho, \theta$ on levels, $w$ on interfaces), and Charney-Phillips (CPH) ($u, v, \rho$ on levels, $w, \theta$ on interfaces) configurations. The two key diagnosed variables, $\Pi$ and $u^\xi$ are co-located with $\rho$ and $w$ respectively. Table 1 provides a reference nomenclature for the various discrete derivative operators that arise in the SNFEM.



When needed, the contravariant $\xi$ velocity is computed on interfaces via

$$u_i^\xi = g^{\xi\alpha}\mathcal{I}_i^n(u_\alpha)_n + g^{\xi\beta}\mathcal{I}_i^n(u_\beta)_n + g^{\xi\xi}|\mathbf{g}_\xi|w, \tag{47}$$

where all contravariant metric terms are evaluated on interfaces.

### 3.3 Temporal Discretization

The equations (6)-(10) are written in the form

$$\frac{\partial\psi}{\partial t} - f(\mathbf{x},\psi) = g(\mathbf{x},\psi), \tag{48}$$

where $f(\mathbf{x},\psi)$ denotes terms associated with non-stiff modes, *i.e.* horizontally-propagating modes and vertical advection of horizontal velocity. The function $g(\mathbf{x},\psi)$ denotes geometrically stiff terms associated with all vertical derivatives except for vertical advection of horizontal velocity. The model follows the approach of Ullrich and Jablonowski (2012) by treating non-stiff terms using an explicit temporal operator and stiff terms using an implicit operator.

For the first time step, an implicit update is applied,

$$\psi^{(0)} = \psi^n + \tfrac{\Delta t}{2}(\mathcal{I} - \tfrac{\Delta t}{2}\mathcal{DG}(\psi^n))^{-1}\mathcal{G}(\psi^n), \tag{49}$$

where $\mathcal{G}(\psi^n)$ represents the discretization described in section 3.2 and $\mathcal{DG}(\psi^n) = \partial\mathcal{G}/\partial\psi^n$. For later time steps, the implicit update is instead obtained from a stored tendency,

$$\psi^{(0)} = \psi^n + \tfrac{\Delta t}{2}\overline{\psi}. \tag{50}$$

Explicit terms are evolved using a Runge-Kutta method which supports a large stability bound for spatial discretizations with purely imaginary eigenvalues. This particular scheme is based

on Kinnmark and Gray (1984a,b) and takes the form

$$\psi^{(1)} = \psi^{(0)} + \tfrac{\Delta t}{5} f(\psi^{(0)}),$$
$$\psi^{(2)} = \psi^{(0)} + \tfrac{\Delta t}{5} f(\psi^{(1)}),$$
$$\psi^{(3)} = \psi^{(0)} + \tfrac{\Delta t}{3} f(\psi^{(2)}), \tag{51}$$
$$\psi^{(4)} = \psi^{(0)} + \tfrac{2\Delta t}{3} f(\psi^{(3)}),$$
$$\psi^{(5)} = -\tfrac{1}{4}\psi^{(0)} + \tfrac{5}{4}\psi^{(1)} + \tfrac{3\Delta t}{4} f(\psi^{(4)}).$$

Hyperviscosity is then applied in accordance with Ullrich (2014a), with scalar hyperviscosity used for all scalar quantities and vector hyperviscosity used for the horizontal velocity field. Mathematically, the update takes the form,

$$\psi_s^{(6)} = \psi_s^{(5)} + \Delta t \mathcal{H}(\nu)\mathcal{H}(1)\psi_s^{(5)}, \tag{52}$$
$$\mathbf{u}^{(6)} = \mathbf{u}^{(5)} + \Delta t \mathcal{H}(\nu_d, \nu_v)\mathcal{H}(1,1)\mathbf{u}^{(5)}, \tag{53}$$

where $\psi_s \in \{\theta, w, \rho\}$.

When active, Rayleigh friction is applied via backward Euler to relax all prognostic variables to a specified reference state,

$$\psi^{(7)} = \gamma \psi^{(6)} + (1-\gamma)\psi^{\text{ref}}, \tag{54}$$

where $\gamma = [1 + \nu_r(\mathbf{x})\Delta t]^{-1}$ is in terms of the Rayleigh friction strength $\nu_r(\mathbf{x})$.

In accordance with Strang splitting, a final implicit update is applied,

$$\overline{\psi} = (\mathcal{I} - \tfrac{\Delta t}{2}\mathcal{D}\mathcal{G}(\psi^{(7)}))^{-1}\mathcal{G}(\psi^{(7)}), \tag{55}$$
$$\psi^{n+1} = \psi^{(7)} + \tfrac{\Delta t}{2}\overline{\psi}. \tag{56}$$

# 4 Validation

In this section we present a set of test cases with the purpose of investigating the performance of the SNFEM for mesoscale atmospheric modeling. Our emphasis is on a wide range of resolu-





tions from the global scale (200 km) to the large eddy scale (5 m). These scales transition from hydrostatic to scales where all non-linear terms in the equations (6) - (10) become significant. For our experiments we will hold the following components of the computations constant:

(1) The horizontal discretization is kept as a standard $4^{th}$ order spectral element formulation for all simulations, as outlined in section 3.1.

(2) The time integration scheme is based on Strang-split IMplicit EXplicit (IMEX) outlined in section 3.3.

(3) Vertical terms $\frac{\partial}{\partial z}$ are integrated implicitly using the generalized minimal residual method (GMRES) with no preconditioner.

(4) Reference solutions employ consistent $4^{th}$ order vertical and horizontal discretizations at a resolution at least $2\times$ finer than the evaluation experiments

(5) Unless stated otherwise, all of the results correspond to Lorenz (LOR) staggering in the vertical.

(6) The total number of vertical levels in each test is kept constant. Only the vertical order of accuracy is changed and consequently the distribution of grid spacing according to the locations of element nodes.

For these tests, we investigate the effect of a relatively high-order $n_{vp} = 10$ vertical coordinate on flow results at resolutions coarser than the reference solutions. Our hypothesis is that flow structures and measures of interest will be better approximated using the high-order discretization. We consider the properties of our arbitrary order methods in the context of meshes with mixed grid resolutions such as static and adaptive variable resolution experiments. A primary benefit of using the higher order SNFEM is improved accuracy even with a coarser vertical grid.

Reference results are computed with a consistent spatial (horizontal and vertical) discretization or order "O4". Experiments done at coarser resolutions with varying vertical order of accuracy are titled "VO#".



## 4.1 Steady-state geostrophically balanced flow in a channel

The first test represents steady-state geostrophically balanced flow in a channel and is based on a new test case defined by Ullrich et al. (2015). The domain is a channel of dimensions $L_x \times L_y \times L_z$ with periodic boundaries in the $x$ direction and no-flux conditions at all other interfaces. In this case we choose $L_x = 30000$ km, $L_y = 6000$ km and $L_z = 30$ km. The shorter zonal width compared with that of Ullrich et al. (2015) was chosen for reasons of computational efficiency and did not affect the final solution. The initial flow is comprised of a zonally-symmetric mid-latitudinal jet, defined so that the wind is zero at the surface and along the y-boundary. Hyperviscosity is applied in the horizontal and vertical at $4^{th}$ order as well as a sponge layer at the top and longitudinal boundaries. The sponge layers are used to prevent the accumulation of standing wave reflections in the flow. This formulation can either be on an $f$-plane or $\beta$-plane, which have Coriolis parameters

$$f = f_0, \quad \text{and} \quad \beta = f_0 + \beta_0(y - y_0), \tag{57}$$

respectively, where $f_0 = 2\Omega \sin\varphi_0$ and $\beta_0 = 2a^{-1}\Omega \cos\varphi_0$ at latitude $\varphi_0 = 45°$N. Here, the radius of the Earth is $a = 6371.229 \times 10^3$ m, its angular velocity is $\Omega = 7.292 \times 10^{-5}$ s$^{-1}$ and $y_0 = L_y/2$ is the center point of the domain in the y-direction.

The simulation is performed for the original $\beta$-plane configuration outlined in Ullrich et al. (2015) where the jet is perturbed directly by a "bump" in the zonal wind that is vertically uniform where $u_p = 1.0$ m s$^{-1}$ centered at $x_c = 2000$ km and $y_c = 2500$ km.

$$u'(x,y) = u_p \exp\left[-\left(\frac{(x-x_c)^2 + (y-y_c)^2}{L_p^2}\right)\right] \tag{58}$$

The grid spacing for the reference solution is $\Delta x = 50$ km, $\Delta y = 50$ km, $\Delta z = 0.75$ km and $\Delta t = 30$ s. Experiments are conducted at vertical order 2 and 10 at a resolution of $\Delta x = 200$ km, $\Delta y = 200$ km, $\Delta z = 1.5$ km and $\Delta t = 240$ s. The $4^{th}$ order scalar and vector (vorticity and





divergence separately) diffusion coefficients in are given by

$$\nu_{\text{scalar}} = 1.0 \times 10^{14} \left( \frac{\Delta x}{L_{\text{ref}}} \right)^{3.2} \text{m}^4 \text{s}^{-1}, \tag{59}$$

$$\nu_{\text{vorticity}} = 1.0 \times 10^{14} \left( \frac{\Delta x}{L_{\text{ref}}} \right)^{3.2} \text{m}^4 \text{s}^{-1}, \tag{60}$$

$$\nu_{\text{divergence}} = 1.0 \times 10^{14} \left( \frac{\Delta x}{L_{\text{ref}}} \right)^{3.2} \text{m}^4 \text{s}^{-1}. \tag{61}$$

where $\Delta x$ is the element length in the $x$ direction and $L_{\text{ref}} = 11.0 \times 10^5$ m is the reference length used for this test case. For this test, vertical flow-dependent viscosity is disabled since it did not have a clear impact on the solution.

The baroclinic instability is a primary mechanism for the development of mid-latitude storm systems and so it is important that an atmospheric modeling platform reproduce these phenomena accurately. We present a reference solution of the baroclinic wave shown in Fig. 3 that is approaching the transition into the non-hydrostatic regime. We are interested in estimates of vertical motion where the reference solution shows maxima on the order of 2 cm s$^{-1}$. Regions of strong vertical motion correspond to strong horizontal gradients in the vorticity and temperature fields and we expect that non-hydrostatic effects will be locally significant.

The reference solution for temperature and vorticity at 500m elevation shown here can be compared at day 10 with the original results from Ullrich et al. (2015) produced with MCore Ullrich and Jablonowski (2012) to verify that Tempest is computing a consistent solution. In particular we expect that vertical motion will be under-predicted in coarser models at a given order of accuracy.

The vorticity field at coarse resolution (Fig. 4) is largely unaffected by changes in vertical order. However, the vertical velocitiy (Fig. 5), and by association the horizontal divergence (not shown), shows a substantial increase in magnitude as order increases. This increase aligns the vertical velocity more closely with the reference solution magnitiude (greater than 1 cm s$^{-1}$) using the $10^{th}$ order vertical coordinate as shown in Fig. 5. We conclude that although the higher





order vertical coordinate does not substantially impact the horizontal character of the solution, it does better capture the magnitude of vertical velocity, particularly in frontal regions. We note that the coarse resolution chosen here is nearly double that of current operational climate modeling systems and well within the hydrostatic regime.

## 4.2 Schär mountain

Atmospheric flows are strongly influenced by the lower boundary, where topographically-induced waves transport momentum and energy vertically. Schär et al. (2002) describes a uniform zonal flow field over orography that leads to the generation of a stationary mountain response, consisting of a linear combination of hydrostatic and non-hydrostatic waves. The atmosphere is initially under uniform stratification with constant Brunt-Väisälä frequency $\mathcal{N} = 0.01$ s$^{-1}$. The temperature and pressure are $p_0 = 1000$ hPa and $T_0 = 280$ K at $z = 0$ m. To trigger the standing waves, an initial uniform mean flow of $\overline{u} = 10$ m s$^{-1}$ is prescribed over the topographic profile given by

$$h_T(x) = h_c \exp\left[-\left(\frac{x}{a_c}\right)^2\right]\cos^2\left(\frac{\pi x}{\lambda}\right), \tag{62}$$

with parameters $h_c = 250$ m, $\lambda = 4000$ m and $a_c = 5000$ m. The simulation domain is $(x, z) \in [-30, 30] \times [0, 25]$ km with a no-flux boundary specified along the bottom surface. Free-flow boundary conditions are prescribed at the top and lateral boundaries with a Rayleigh sponge layer 10 km wide along the lateral boundaries and 10 km deep at the model top. Note that the domain bounds differ from Schär et al. (2002) to minimize the effect of the Rayleigh layers on the flow interior. The simulation is run to $t = 10$ h, when the solution has reached a quasi-steady state. For these simulations, no explicit dissipation is applied in either the horizontal or vertical and Coriolis forcing is set to zero throughout.

To validate that Tempest produces the correct mountain wave response, the Schär mountain test was performed until $t = 10$h with a relatively fine resolution of $\Delta x = 100$ m, $\Delta z = 100$ m and $\Delta t = 0.2$ s. For the Schär test case results, we present the LOR configuration only as CPH





shows no apparent difference for these flow conditions. As shown in Fig. 6 (left) Tempest accurately reproduces the vertical velocity field at the reference resolution (for comparison with another numerically derived solution, see Giraldo and Restelli (2008)). We also show the analytical solution based on linear mountain theory following Klemp et al. (2003); Smith (1979) overlaid in dotted contours. As pointed out by Klemp et al. (2003), an inconsistent treatment of the topographic metric terms in this formulation can lead to the generation of spurious waves which is not observed in this case.

As discussed in Thuburn and Woollings (2005) and Thuburn (2006), staggering is necessary for eliminating stationary computational modes that arise in collocated discretizations. To better understand the impact of staggering, Fig. 7 demonstrates the use of the collocated or unstaggered configuration which shows a highly-oscillatory stationary mode that pollutes the solution relative to the Lorenz configuration at the same resolution. The plots show errors in the vertical velocity near the bottom boundary condition and errors throughout the flow field due to the vertical mode. This artifact is conspicuously absent from both LOR and CPH runs.

Experiments are conducted at vertical order 2, 4, 10, and 40 (in the limit where the polynomial order is equal to the total number of levels, denoted ST) at a relatively coarse resolution of $\Delta x = 500$ m, $\Delta z = 500$ m and $\Delta t = 0.4$ s. Results are depicted in Fig. 8 and the difference against the reference solution in Fig. 9. The $2^{nd}$ order results show substantial disagreement with the reference solution that is enhanced at altitude. This result appears to be associated with an overestimation of the vertical wavelength of the mountain response that arises from the lower order discretization. At $4^{th}$ order the upper atmosphere does not show substantial errors, and most differences are instead constrained to the near-surface. These near-surface errors generally show consistent improvement as the vertical order-of-accuracy is increased. The discrepancy that appears at the highest peak of the Schär mountain ($x = 0$) is associated with slight differences in resolving the topography at coarser horizontal resolution than the reference solution.

We further compare the resulting profiles of momentum flux for all experiments in the Lorenz configuration (Fig. 10). We observe that the flux profile for the $2^{nd}$-order method has the greatest error, as expected from dispersion errors typical of low order centered schemes (particularly



in the upper atmosphere and near the surface). The higher-order methods show improvements in the structure (especially near the surface) and magnitude of the profiles, but again appear to be influenced by the lower order horizontal discretization. Furthermore, the results are strongly influenced by the Rayleigh layer showing a pronounced deviation in the flux profiles through-out the domain. The Rayleigh layer approximation to a free-flow boundary condition clearly introduces deficiencies that are exacerbated in the flux provides.

## 4.3 Straka density current

The density current test case of Straka et al. (1993) considers a cold bubble that sinks and spreads across the bottom boundary, driving the development of Kelvin-Helmholtz rotors. The original experiments by Straka et al. (1993) sought a converged solution through the use of $2^{nd}$ order uniform diffusion applied to all prognostic variables. A value of $\nu = 75$ m$^2$ s$^{-1}$ was chosen so that a horizontal resolution of $\Delta x = 25$ m was sufficient for convergence. No-flux conditions are applied on all boundaries and Coriolis forcing set to zero.

The initial state consists of a hydrostatically balanced state with a uniform potential temperature of $\theta = 300$ K. A standard pressure of $p_0 = 1000$ hPa is assumed. The cold bubble perturbation is applied to the $\theta$ field and is given by

$$\theta' = \begin{cases} 0 & \text{if } r > 1.0, \\ -\frac{\theta_c}{2}\left[1 + \cos\left(\pi r\right)\right] & \text{if } r \leq 1.0, \end{cases} \tag{63}$$

where $\theta_c = -15$ K and

$$r = \sqrt{\left(\frac{x - x_c}{x_r}\right)^2 + \left(\frac{z - z_c}{z_r}\right)^2}. \tag{64}$$

The domain is an enclosed box $(x, z) \in [-25600, 25600] \times [0, 6400]$ m with $t \in [0, 900]$ s. The cold bubble is initially located at $(x_c, z_c) = (0, 3000)$ m with radius $(x_r, z_r) = (4000, 2000)$ m.





The $4^{th}$ order horizontal hyperdiffusion coefficients for all fields are given by

$$\nu_{\text{scalar}} = 5.0 \times 10^{12} \left(\frac{\Delta x}{L_{\text{ref}}}\right)^{3.2} \text{m}^4\text{s}^{-1}, \tag{65}$$

$$\nu_{\text{vorticity}} = 2.0 \times 10^{14} \left(\frac{\Delta x}{L_{\text{ref}}}\right)^{3.2} \text{m}^4\text{s}^{-1}, \tag{66}$$

$$\nu_{\text{divergence}} = 2.0 \times 10^{14} \left(\frac{\Delta x}{L_{\text{ref}}}\right)^{3.2} \text{m}^4\text{s}^{-1}, \tag{67}$$

where $\Delta x$ is the element length in the $x$ direction and $L_{\text{ref}} = 51200.0$ m is the reference length used for this test case.

For the experiments with vertical flow-dependent hyperviscosity, the viscous coefficients are given by (46). The uniform Laplacian diffusion requires further stabilization via the addition of $4^{th}$ order scalar hyperviscosity in the horizontal and $4^{th}$ order vertical flow-dependent diffusion on all variables. This added diffusivity is necessary to control a horizontal stationary mode in the scalar fields and fast moving vertical modes that are a consequence of sound waves accumulating energy at the grid scale. However, the highly scale-selective nature of the high degree operators does not significantly affect the structure of the reference solution as shown in Fig. 11.

The grid spacing for the reference solution is $\Delta x = 25$ m and $\Delta z = 25$ m with $\Delta t = 0.01$ s. Experiments are further conducted at vertical order 2 and 10 at a resolution of $\Delta x = 200$ m and $\Delta z = 200$ m with $\Delta t = 0.01$ s.

For the density current, we emphasize results from the Lorenz (LOR) staggering. Since only $w$ is specified at the lower boundary, we can implement the correct kinematic condition without triggering a further constraint on the vertical transport of potential temperature (computed on levels) near the boundary. In contrast, since vertical transport of $\theta$ emerges from the term $u^\xi \partial\theta/\partial\xi$, the Charney-Phillips (CPH) configuration effectively blocks transport into the interfacial layer at the bottom boundary and so generates strong vertical gradients of $\theta$ near the surface. These gradients then enhance vertical heat fluxes above the surface, slowing the propagating cold pool as momentum is transported vertically. This inconsistency is counteracted by



the application of uniform diffusion, which provides a mechanism by which $\theta$ can be exchanged with the bottom interface. However, flow-dependent vertical diffusion, which is weighted by $|u^\xi|$, does not permit exchange with the interface and so leads to inconsistency between the LOR and CPH staggerings. In Fig. 11, the CPH staggering with flow-dependent diffusion leads to a relatively slow density current that is more convective near the boundary. Nonetheless, a better choice of flow-dependent coefficient could be made to mitigate this issue.

In practice, we often desire diffusion to be as weak as possible while still preserving the stability of the underlying method. However, as can be seen here, the structure of the density current is also strongly dependent on the dissipation mechanisms employed in the simulation. Here we present the reference solution equivalent to Straka et al. (1993) at the converged resolution. We also compare solutions with different diffusion mechanisms in Fig. 11. In Table 2 it is apparent the reference solutions are sensitive to diffusion and differ significantly in structure, but the wave front positions compare with good precision to the solution given by Straka et al. (1993). This would indicate that momentum fluxes are comparable, but close inspection of the eddy structure suggests significant differences exist throughout, as noted above, and with the appearance of detached eddies when the high-order flow-dependent viscosity is used exclusively.

From Table 2 it is apparent our coarse-resolution experimental solutions are slow with reference damping and $2^{nd}$ order flow-dependent viscosity, but are closer to the reference solution with $4^{th}$ order diffusion. Both low- and high-order simulations show wave front positions that accurately approximate the reference results. However, the structure of the Kelvin-Helmholtz rotors changes significantly with vertical order-of-accuracy and dissipation method shown in Fig. 12. The more scale-selective $4^{th}$ order flow-dependent viscosity shows greater detail in the structure of the rotors. In general, it is not recommended to use hyperdiffusion with a higher order than the dynamical discretization (bottom left) since the impact of the hyperdiffusion will be in the truncation order of the method.

Curiously, the $10^{th}$ order vertical discretization with $4^{th}$ order flow-dependent viscosity produces a flow that more closely approximates the reference solutions at a resolution that would otherwise be considered too poor for the dynamical features considered. However, the authors





have not found a dynamical reason for correlation involving high-order dissipation schemes and the reference solution with uniform damping. Wave front position at the $-1.0\,^{\circ}C$ contour further given in 2 confirm that momentum fluxes are also captured more accurately as these are associated to the propagation speed of the wave front.

## 4.4 Rising thermal bubble

Thermal bubble experiments have become a standard in the development of non-hydrostatic mesoscale modeling systems. At very fine resolutions ($<$ 10 m) we test the ability to reproduce the simplest form of convection. This is a precursor to simulations of real atmospheric phenomena such as thunderstorms and other convective systems. A positive, symmetric perturbation to the potential temperature (buoyancy imbalance) causes a vertical acceleration that moves the bubble upward. Subsequently, shearing and compensating subsidence leads to two primary symmetrical eddies that further break down as the simulation progresses. We are interested in the evolution of the flow in terms of structure and conservative properties on $\theta$.

We present two flow scenarios: a) the bubble rises and is allowed to interact with the top and lateral boundaries and b) the so-called Robert smooth bubble experiment (as outlined in Giraldo and Restelli (2008)) that are a variation of the experiments of Robert (1993). In the former, the bubble will meet the boundaries and develop shear instabilities and in the Robert bubble, shear instabilities develop in the interior of the flow. For these experiments, $4^{th}$ order viscosity is applied in the horizontal and vertical primarily to the potential temperature field. Furthermore, at finer resolutions we observe more fine-scale features of the thermal bubble, including tighter winding of the trailing edges at later times and sharper spatial gradients. Nonetheless, our comparisons for this test case are purely qualitative but remain consistent with previous results.

The background consists of a constant potential temperature field $\overline{\theta} = 300$ K, with a small perturbation of the form

$$\theta' = \begin{cases} 0 & \text{for } r > r_c, \\ \frac{\theta_c}{2}\left[1 + \cos\left(\frac{\pi r}{r_c}\right)\right] & \text{for } r \leq r_c, \end{cases} \tag{68}$$





where

$$r = \sqrt{(x - x_c)^2 + (z - z_c)^2}. \tag{69}$$

Here we choose the amplitude and radius of the perturbation to be $\theta_c = 0.5$ K and $r_c = 250$ m, respectively. The domain consists of a rectangular region $(x, z) \in [0, 1000] \times [0, 1000]$ m for the thermal bubble and $(x, z) \in [0, 1000] \times [0, 1500]$ m for the Robert bubble with $t \in [0, 1200]$ s. The center-point of the bubble is located at $x_c = 500$ m and $z_c = 350$ m for the thermal bubble and $z_c = 260$ m for the Robert bubble. The boundary conditions are no-flux over all boundaries. No sponge layer is used, and Coriolis forces are set to zero.

The reference grid spacing is $\Delta x = 5$ m and $\Delta z = 5$ m respectively. This is considered the reference resolution following Giraldo and Restelli (2008). Experiments are conducted at a relatively coarser resolution of $\Delta x = 10$ m and $\Delta z = 10$ m. The $4^{th}$ order scalar and vector (vorticity and divergence separately) diffusion coefficients in are given by

$$\nu_{\text{scalar}} = 1.0 \times 10^6 \text{ m}^4\text{s}^{-1}, \quad \nu_{\text{vorticity}} = 1.0 \times 10^6 \text{ m}^4\text{s}^{-1}, \quad \nu_{\text{divergence}} = 1.0 \times 10^6 \text{ m}^4\text{s}^{-1}. \tag{70}$$

The $4^{th}$ order scalar and vector (vorticity and divergence separately) diffusion coefficients in are given by

$$\nu_{\text{scalar}} = 1.0 \times 10^6 \left( \frac{\Delta x}{L_{\text{ref}}} \right)^{3.2} \text{m}^4\text{s}^{-1}, \tag{71}$$

$$\nu_{\text{vorticity}} = 1.0 \times 10^6 \left( \frac{\Delta x}{L_{\text{ref}}} \right)^{3.2} \text{m}^4\text{s}^{-1}, \tag{72}$$

$$\nu_{\text{divergence}} = 1.0 \times 10^6 \left( \frac{\Delta x}{L_{\text{ref}}} \right)^{3.2} \text{m}^4\text{s}^{-1}. \tag{73}$$

where $\Delta x$ is the element length in the $x$ direction and $L_{\text{ref}} = 1000.0$ m is the reference length used for this test case.

Rising bubble experiments show the non-linear dynamics of dry 2D convection. The classic thermal bubble test shown in Fig. 13 shows potential temperature being advected conservatively




throughout the domain at the reference resolution. These results use a dissipation mechanism that combines $4^{th}$ order hyperdiffusion of $\theta$ for horizontal modes and scale-adaptive $4^{th}$ order flow-dependent hyperviscosity of $\theta$ for vertical modes. In this case, no diffusion is needed in the velocity or density fields to obtain a stable simulation.

The rising thermal bubble experiment is typically carried out and compared at 700 seconds precisely before the convective bubble interacts with the top boundary of the domain. We present this result for comparison with previous results in Fig. 15. However, it is also important to evaluate the conservative properties of the method and we carry out the simulation to 1200 seconds. Since (9) is a strict statement of constant potential temperature following fluid parcels, the results of Fig. 15 compared to Fig. 13 demonstrate that our method is stable and approximates conservation of $\theta$ closely when a high-order vertical discretization is used.

The Robert smooth bubble experiment extends the vertical domain allowing for the onset of Kelvin-Helmholtz instabilities in the flow. The solution at the reference resolution is shown in Fig. 14. The exact time and manner in which the instabilities arise is strongly dependent on the vertical order and dissipation method used in the simulation. In the reference solution, the onset of unstable eddies begins at approximately 900 s with the flow transitioning into vigorous mixing in the region of the primary rotors.

High-order vertical discretizations are typically associated with strong oscillations that can induce perturbations that grow into unstable eddies. The net effect is that a high-order vertical discretization, given the same horizontal discretization, changes the local mixing characteristics of the flow. This effect is seen clearly in Fig. 16. The $10^{th}$ order simulation has a structure that more closely approximates the reference result in Fig. 14. In the context of studies that seek to represent convective processes, we would expect entrainment fluxes to be improved at a coarser resolution with the higher-order vertical discretizations.





# 5   Conclusions

The idea of separating the vertical and horizontal dynamics in atmospheric modeling systems has roots in the scale differences that characterize atmospheric flows. This principle has been fully exploited in the development of global and mesoscale models, along with the applica-
tion of the hydrostatic approximation. This paper adds to the modern literature on modeling atmospheric dynamics by analyzing a novel discretization technique for achieving high-order accuracy in the vertical while maintaining the desirable properties of staggered methods. We refer to this technique as the Staggered Nodal Finite Element Method (SNFEM).

The test suite we present in this work is not exhaustive, but it is intended to evaluate the performance of the numerical schemes under conditions of near hydrostatic synoptic scale flow in section 4.1, linear, mesoscale, non-hydrostatic flow with topography in section 4.2, and fully nonlinear, non-hydrostatic, Large Eddy Simulation (LES) scale, flow in section 4.3 and section 4.4. As global models progress into into the regime of non-hydrostatic flows, real flow cases will be characterized by one or more of the properties mentioned, and likely in combination when variable or adaptive meshing methods are used. More importantly, we expect that uniform or mixed grids being prepared in research will begin to span the scale range that includes the transition to non-hydrostatic dynamics and on to large-eddy flows.

In general, we postulate that a higher-order method based on finite elements will be more accurate at a given resolution with minimal computational cost relative to a low order method. Our results demonstrate that a high-order vertical coordinate approximates well resolved, refer-
ence results at coarser resolutions that would be otherwise considered poorly represented. Our experiments nonetheless are constrained by the order of horizontal and temporal discretizations. Therefore, we restrict our recommendation to the use of $4^{th}$ order SNFEM as optimum for the tests given here. In general the combined spatial order of accuracy should be consistent to max-
imize the effect of increased accuracy. The high-order approximation provides an improvement to the vertical dynamics and so reduces the need for higher vertical resolution. This benefit would prove effective when variable-grid methods are considered and nesting mesh levels can be saved by employing the SNFEM at high-order. The use of staggering in conjunction with





high-order has further benefits, in particular the avoidance of stationary computational modes that are known to persist with co-located methods.

However, there are some trade offs when increasing the vertical order: 1) for a vertically implicit method, fewer high-order elements lead to a dense matrix structure that is more expensive to invert, 2) the oscillatory nature of the polynomial functions that make up the interpolants within an element have physical consequences (involving nonlinear processes) at the smallest scales, and 3) higher-order spatial discretizations often require smaller time steps or higher order temporal discretizations. Fig. 3 shows the times required for computations of varying vertical order and processor scaling. The results confirm that the relative cost in moving to $4^{th}$ order is indeed modest relative to the use of higher orders. The first point can be addressed in the construction of the software where parallelization and correct use of hardware resources minimizes the dense operations that high-order elements imply. We saw in Fig. 16 that oscillations associated with high-order interpolants helped to approximate fine scale structures, but these oscillations can also be harmful depending on the flow condition. While vertical order of accuracy can be increased up to the total number of vertical levels, e.g. results from the Schär cases in Fig. 8, increasing computational expense indicates that intermediate orders of accuracy will generally be most effective. In this study, many of the results at $4^{th}$ order sufficiently improve solutions relative to low order alternatives.

Furthermore, when physical instabilities arise, a consistent, high-order, and scale selective dissipation strategy is necessary. In this regard, finite element methods allow for the construction of diffusion operators for this purpose e.g. section 3.2.3.We can experiment with different combinations of diffusion operators including coefficients that are variable in space. While scale-selective $4^{th}$ order operators with some grid resoltuion dependence are sufficient for this work, we intend to explore a wider range of strategies based on polynomial filtering, variational multiscale methods, etc. with the goal of eliminating the tuning procedure associated with user-provided coefficients.

The numerical dissipation strategy implemented here serves two primary goals: 1) stabilization of the computations and 2) as a form of closure for the Euler equations solved on a truncated grid. The methods we employ allow for the construction of derivative operators of various or-





ders in a consistent manner. Tempest features a system that allows for diffusion to be applied in a selective manner on variables that is split according to the time integration scheme.

Further experiments are necessary to test the extent of the third point above. For this work, we used a $2^{nd}$ order Strang time integration scheme (section 3.3) that was sufficiently robust to carry out all of the experiments up to $10^{th}$ order without overly restricting time step size relative to the $2^{nd}$ order simulations.

The authors conclude the following based on the experiments conducted and properties of the SNFEM:

1. Staggering has been generalized to finite element methods combining continuous and discontinuous element formalisms. The result is a method that closely parallels the behavior of staggered finite differences eliminating stationary modes.

2. Variable order of accuracy is an effective strategy that can compensate for limitations in grid scale resolution. However, the effects at very high order must be understood and controlled with appropriate stabilization methods. In general, "intermediate" orders (about $4^{th}$ order) are recommended with consideration for consistency in overall spatial order given an IMEX partitioned architecture

We note that, while the equations are formulated in a coordinate-free manner, the results given all correspond to regular Cartesian coordinates. Experiments corresponding to small planet and global domains are left for a subsequent work. However, any curved geometry with a terrain-following surface topography can be applied to the equations since all grid information is held in the metric terms described in section 2. As such, the effects of curved geometry and variable vertical order-of-accuracy are only addressed here in the Schär and Baroclinic wave cases (using the $\beta$ plane approximation). From a design perspective, metric terms are precomputed and derivative operators are built in the natural, local coordinate frame when any grid is used.



# 6   Code and/or data availability

The Tempest codebase used to generate the results in this publication are available through the following Git repository: https://github.com/paullric/tempestmodel.

*Acknowledgements.*  The authors would like to thank Dr. Hans Johansen, Dr. Mark Taylor and Dr. David Hall for their assistance in refining this manuscript. Funding for this project has been provided by the Department of Energy, Office of Science project "A Non-hydrostatic Variable Resolution Atmospheric Model in ACME."





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





| Variable | Term | Choice of Staggering | | |
|---|---|---|---|---|
| | | SE $(\rho_i, \theta_i, w_i)$ | SNFEM-LOR $(\rho_n, \theta_n, w_i)$ | SNFEM-ChP $(\rho_n, \theta_i, w_i)$ |
| $u, v$ | $\Pi$ | $\Pi_i(\rho_i, \theta_i)$ | $\Pi_n(\rho_n, \theta_n)$ | $\Pi_n(\rho_n, \mathcal{I}_n^i \theta_i)$ |
| $\theta$ | $u^\xi \dfrac{\partial \theta}{\partial \xi}$ | $(u_i^\xi)\mathcal{D}_i^i\theta_i$ | $(\mathcal{I}_n^i u_i^\xi)(\mathcal{D}_n^n\theta)$ | $(u_i^\xi)(\mathcal{D}_i^i\theta_i)$ |
| $w$ | $\theta \dfrac{\partial \Pi}{\partial \xi}$ | $\theta_i \mathcal{D}_i^i \Pi_i$ | $(\mathcal{I}_n^i\theta_n)(\mathcal{D}_i^n\Pi_n)$ | $\theta_i(\mathcal{D}_i^n\Pi_n)$ |
| $\rho$ | $\dfrac{1}{J}\dfrac{\partial}{\partial \xi}(J\rho u^\xi)$ | $\dfrac{1}{J_i}\mathcal{D}_i^i(J_i\rho_i u_i^\xi)$ | $\dfrac{1}{J_n}\mathcal{D}_n^i\left[J_i(\mathcal{I}_i^n\rho_n)u_i^\xi\right]$ | $\dfrac{1}{J_n}\mathcal{D}_n^i\left[J_i(\mathcal{I}_i^n\rho_n)u_i^\xi\right]$ |

**Table 1.** Composition of interpolation $\mathcal{I}$ and differentiation $\mathcal{D}$ operators for several choices of staggering, including co-located spectral elements (SE), SNFEM with Lorenz staggering (SNFEM-LOR) and SNFEM with Charney-Phillips staggering (SNFEM-ChP). Script $i$ denotes variables defined on interfaces (Gauss-Lobatto nodes) and $n$ represents variables defined on model levels (Gauss nodes). For operator $\mathcal{I}$ and $\mathcal{D}$, the subscript denotes the target ($i$ or $n$) and the superscript denotes the origin.





| Method-Stagger | Vertical Order @ Resolution | Diffusion Method | Wave Front (km) |
|---|---|---|---|
| SNFEM-LOR | 2 @ $\Delta x = 190$ m | Reference Damping | 14.21 |
| SNFEM-LOR | 2 @ $\Delta x = 190$ m | Up-wind Order 2 | 14.59 |
| SNFEM-LOR | 2 @ $\Delta x = 190$ m | Up-wind Order 4 | 15.68 |
| SNFEM-LOR | 4 @ $\Delta x = 190$ m | Reference Damping | 14.18 |
| SNFEM-LOR | 4 @ $\Delta x = 190$ m | Up-wind Order 2 | 14.58 |
| SNFEM-LOR | 4 @ $\Delta x = 190$ m | Up-wind Order 4 | 15.47 |
| SNFEM-LOR | 10 @ $\Delta x = 190$ m | Reference Damping | 14.22 |
| SNFEM-LOR | 10 @ $\Delta x = 190$ m | Up-wind Order 2 | 14.61 |
| SNFEM-LOR | 10 @ $\Delta x = 190$ m | Up-wind Order 4 | 15.33 |
| **FD-Colocated** | **2 REFC @ $\Delta x = 25$ m** | **Explicit $\nu_0 = 75$ m$^2$s$^{-1}$** | **15.53** |
| **SNFEM-LOR** | **4 (REF) @ $\Delta x = 25$ m** | **Reference Damping** | **15.20** |
| SNFEM-LOR | 4 (REF) @ $\Delta x = 25$ m | Up-wind Order 2 | 15.77 |
| SNFEM-LOR | 4 (REF) @ $\Delta x = 25$ m | Up-wind Order 4 | 15.68 |

**Table 2.** Cold wave front position ($\theta' = -1.0$ K) for all orders of accuracy and diffusion methods. Reference damping is uniform $2^{nd}$ order diffusion on all prognostic variables such that $\nu = 75$ m$^2$s$^{-1}$ combined with horizontal hyperdiffusion on scalars and vertical $4^{th}$ order up-wind diffusion. The reference solution wave front position (finite difference method at 25 meter resolution) by Straka et al. (1993) is shown in bold (REFC) compared to the equivalent result from Tempest.





| Vertical Order | # Cores | | |
|---|---|---|---|
| | 1 | 2 | 4 |
| $2^{nd}$ | 0.117 s | 0.070 s | 0.061 s |
| $4^{th}$ | 0.163 s | 0.102 s | 0.082 s |
| $8^{th}$ | 0.248 s | 0.143 s | 0.106 s |

**Table 3.** Thermal bubble test ($\Delta x = 20$ m) average processor time taken per time step in seconds. Intel Core i7 4000 series under Linux with 4 compuational cores on die (no interconnect hardware present). Results show relative scalability for Tempest using MPI architecture and IMEX partitioning with variable vertical order of accuracy. The implicit equations are solved using the GMRES with no preconditioner.





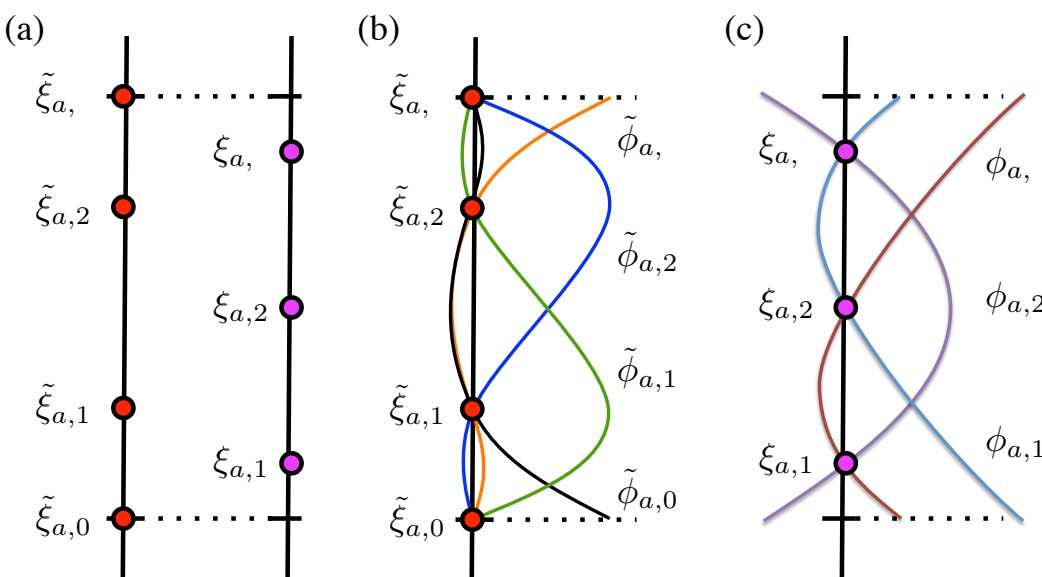

**Fig. 1.** (a) Vertical placement of (left) Gauss-Lobatto nodes and (right) Gauss nodes within a vertical element with $v_{np} = 3$. (b) Basis functions $\tilde{\phi}_{a,k}$ for Gauss-Lobatto nodes within element $a$. (c) Basis functions $\phi_{a,k}$ for Gauss nodes within element $a$.





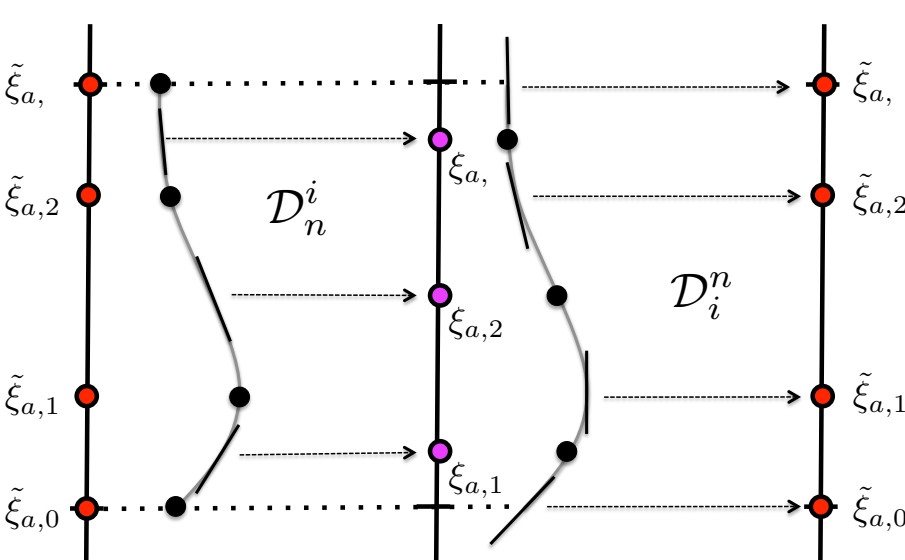

**Fig. 2.** A depiction of the derivative operators $\mathcal{D}_n^i$ and $\mathcal{D}_i^n$, which remap from interfaces to levels and levels to interfaces, respectively. The gray line depicts a typical field variable within element $a$ that emerges from the expansion (left) (23) or (center) (22).



**Fig. 3.** Baroclinic wave in a 3D Cartesian channel at the reference resolution $\Delta x = 100$ km, $\Delta y = 100$ km, $\Delta z = 1$ km at VO4. From top to bottom, temperature, vorticity, vertical velocity, and divergence are shown at day 10 (left) and 15 (right) and at an elevation of 500 m





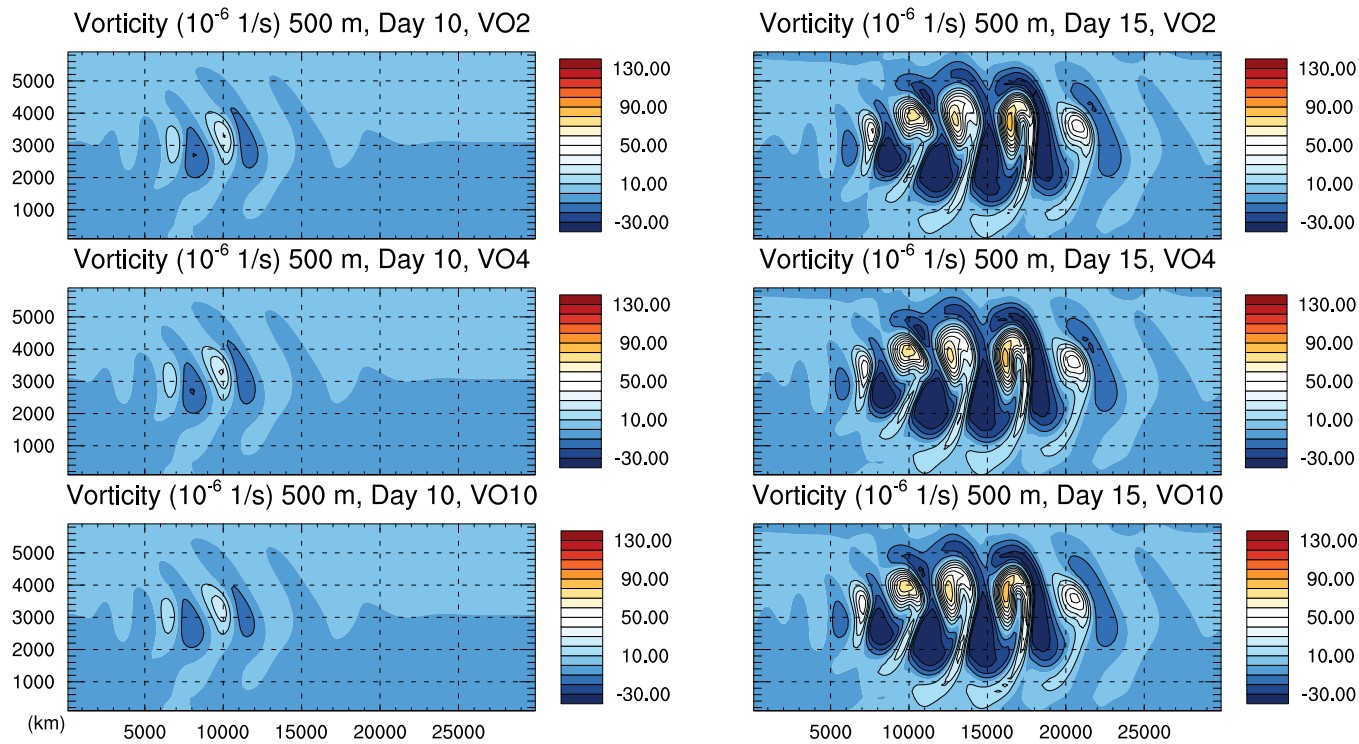

**Fig. 4.** Baroclinic wave in a Cartesian channel at vertical orders 2, 4 and 10. Vorticity at 500 meters on days 10 and 15 at the resolution $\Delta x = 200$ km, $\Delta y = 200$ km, $\Delta z = 1.5$ km.





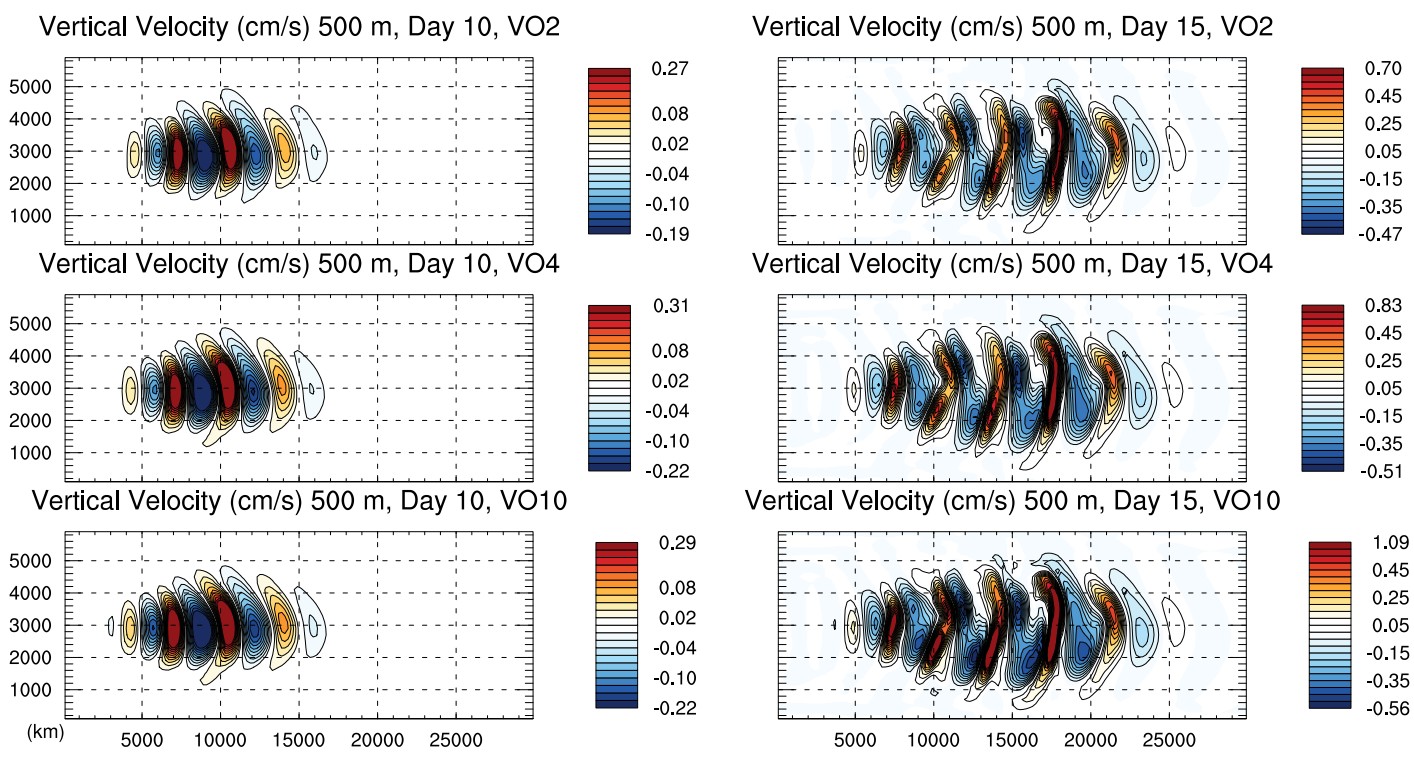

**Fig. 5.** Baroclinic wave in a Cartesian channel at vertical orders 2, 4 and 10. Vertical Velocity at 500m on days 10 and 15 at the resolution $\Delta x = 200$ km, $\Delta y = 200$ km, $\Delta z = 1.5$ km.





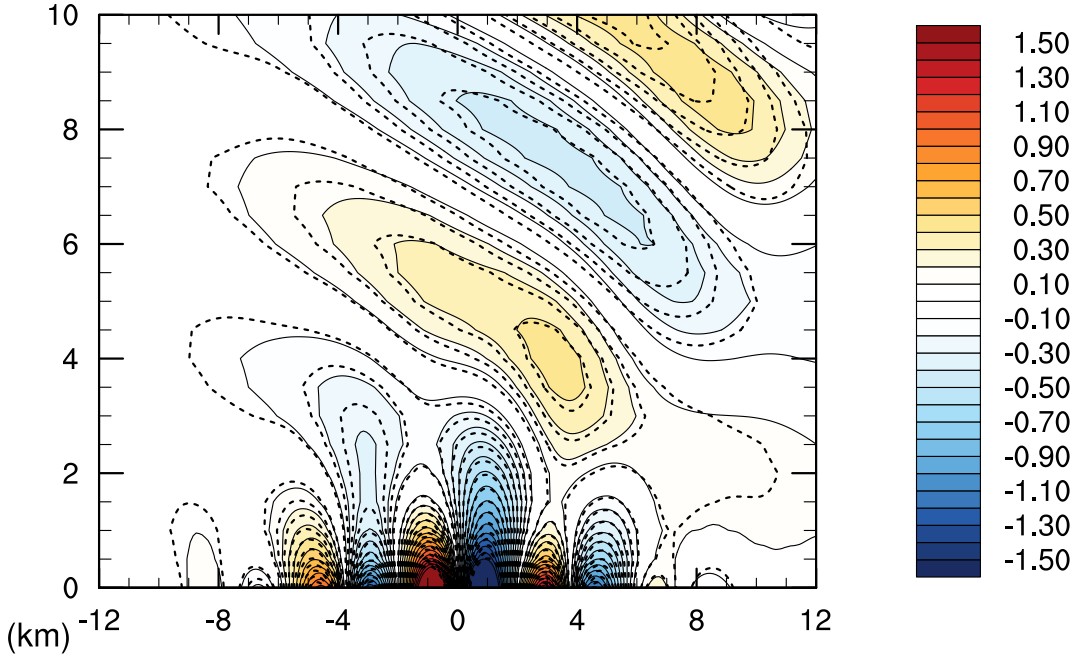

**Fig. 6.** Schär flow at steady state (10 hours) vertical velocity in (m/s) at VO4. Reference resolution shown compared to the analytical solution (dotted contours) from linear mountain wave theory. $\Delta x = 100$ m and $\Delta z = 100$ m.





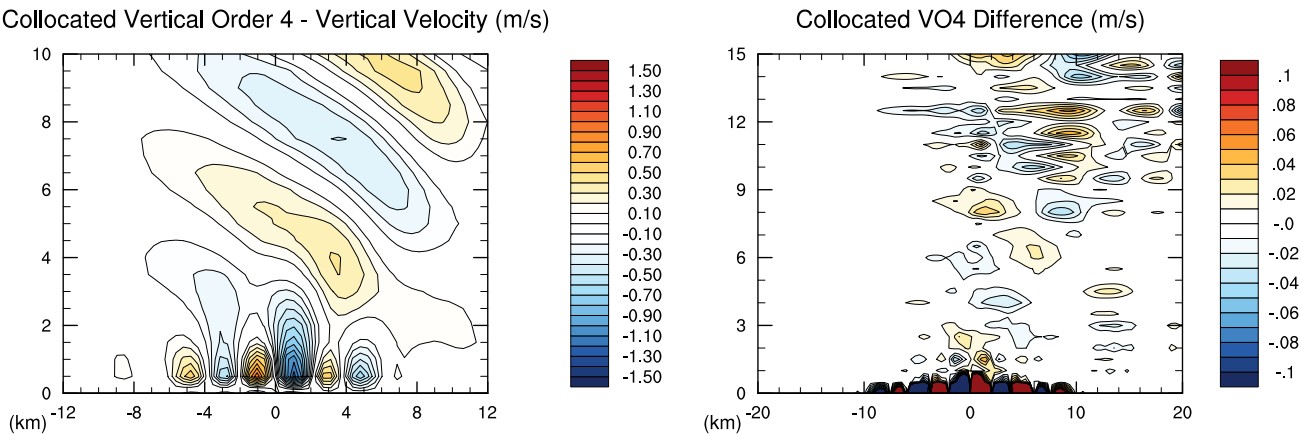

**Fig. 7.** Schär flow at steady state (10 hours). Collocated method (all variables on column levels) result compared to staggered (Lorenz) solution at the same spatial order and resolution. Errors near the terrain boundary and computational modes are apparent in the difference with respect to the staggered solution at the same resolution. $\Delta x = 200$ m and $\Delta z = 200$ m.





**Fig. 8.** Schär flow at steady state (10 hours) vertical velocity in (m/s) at various vertical orders of accuracy (2, 4, 10, and ST) where "ST" stands for single column element spectral transform ($n_{ve} = 1$) with Lorenz (LOR) vertical staggering. Colored contours from Tempest compared to dotted contours for the analytical solution. $\Delta x = 500$ m and $\Delta z = 500$ m.







**Fig. 9.** Schär flow steady state (10 hours). Vertical velocity difference with respect to the reference solution (Fig. 6, left). Results are interpolated to a regular $z$ coordinate with $\Delta z = 500$ m in experiments and reference solution for differencing. Computations performed at $\Delta x = 500$ m and $\Delta z = 500$ m.





**Fig. 10.** Schär mountain vertical profile of momentum flux for all experiments. The flux profiles are computed by $\int_{-X}^{X} \{[\bar{\rho} + \rho'] [\bar{u} + u'] w'\} dx$ at $t = 10$ hours where overbars indicate initial condition values and primes are departures thereof. Results are interpolated to a regular $z$ coordinate with $\Delta z = 500$ m in experiments and reference solution to compute the integral flux. Results are normalized to the value at the surface in the reference solution.





**Fig. 11.** Straka Density Current test reference solutions at vertical order 4 in two staggering configurations LOR and CPH. Converged resolution of $\Delta x = 25$ m and $\Delta z = 25$ m shown. Vertical flow dependent diffusion in of order 2 and 4 (rows 2 and 3) is compared with the reference solution where an explicit $2^{nd}$ order diffusion with $\nu_0 = 75$ m$^2$s$^{-1}$ is used (top row).





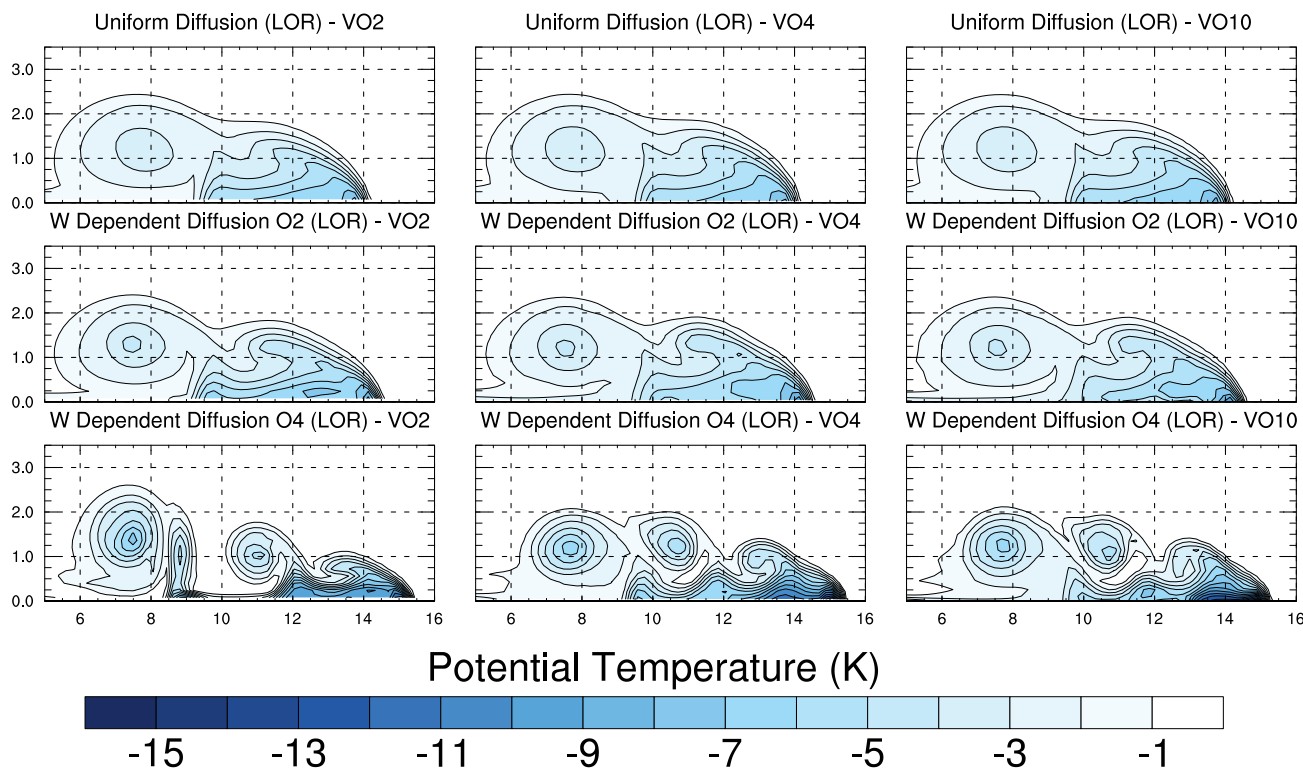

**Fig. 12.** Straka Density Current test at vertical order 2, 4 and 10. Coarse, evaluation resolution of $\Delta x = 190$ m and $\Delta z = 160$ m shown. Vertical flow dependent diffusion of order 2 and 4 (rows 2 and 3) is compared with the reference solution where an explicit $2^{nd}$ order diffusion with $\nu_0 = 75$ m$^2$s$^{-1}$ is used (top row). Results for Lorenz (LOR) staggering shown.





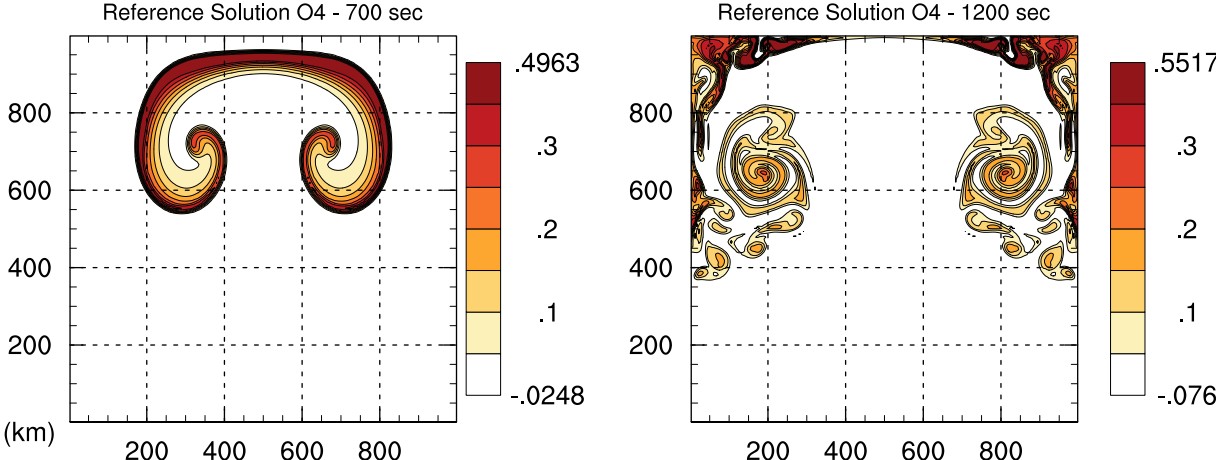

**Fig. 13.** Rising thermal bubble potential temperature reference solution at vertical order 4. Reference resolution $\Delta x = 5$ m and $\Delta z = 5$ m. Flow at 700 and 1200 seconds.

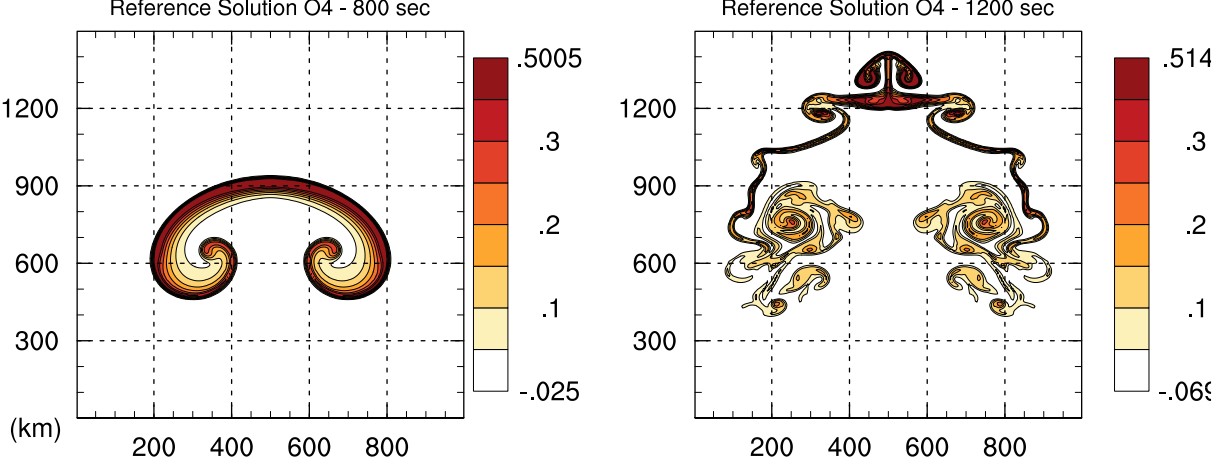

**Fig. 14.** Rising Robert bubble potential temperature reference solution at vertical order 4. Reference resolution $\Delta x = 5$ m and $\Delta z = 5$ m. Flow at 800 and 1200 seconds.





**Fig. 15.** Rising thermal bubble potential temperature at vertical orders 2, 4 and 10. Convection bubbles at 700 and 1200 seconds. Coarse, resolution $\Delta x = 10$ m and $\Delta z = 10$ m. Extrema in $\theta$ shown.



**Fig. 16.** Rising Robert bubble potential temperature at vertical orders 2, 4 and 10. Convection bubbles at 800 at 1200 seconds. Coarse, evaluation resoution $\Delta x = 10$ m and $\Delta z = 10$ m. Extrema in $\theta$ shown.