# Peer review of "Manuscript prepared for Geosci. Model Dev. Discuss. with version 4.1 of the LATEX class copernicus\_discussions.cls. Date: 10 May 2016"

_Geoscientific Model Development, 2015_

## Referee Comment (RC1) · C. Cotter (Referee) · 10 Feb 2016

In this paper, the authors introduce higher-order extensions to vertically staggered grids by using a spectral element approach which was originally introduced in the shallow-water context by Ullrich in another GMD paper. This is very useful and important work, for example in the Gung Ho project we are also considering finite element versions of vertically staggered grids along these lines (although we are currently considering a fully Galerkin weak-form approach) and these initial explorations are very useful to us. I also understand that there are ongoing questions about how to treat the vertical coordinate in the non-hydrostatic version of IFS at ECMWF, since the higher-order splines approach does not work for non hydrostatic IFS and so the standard Lorenz grid

is used. Hence, these ideas have the potential for plenty of impact at these operational centres and elsewhere.

I strongly recommend this paper for publication, but would like the authors to consider a few points to address.

1) around line 18, page 3. The statement about resolving boundary layers troubles me. In a Galerkin finite element setting, the results are basis independent, so the clustering of node points makes no difference. In a spectral element approach, incomplete quadrature is used, so it is not so clear if it helps or not. It would be a shame if the non-specialist reader came away with the impression that the clustering matters in all finite element settings, perhaps it might be better to remove this remark unless the authors can come up with some demonstration or reasoning why it would actually help.

2) p4, l18: perhaps emphasise that you make the comparison at the same DOF count, i.e. lowering p leads to more elements.

3) For structured in the vertical, there is probably not such a big issue here, but for higher p with the same number of elements you get denser blocks in matrices. A naive implementation would see a big increase in cost here, but careful design of kernels that encourage compiler vectorisation can offset this, do you address this or observe any of these effects?

4) section 3.2 I think this could be explained better to the non-expert. I would suggest to first emphasise the split between continuous and discontinuous expansions and then introduce the Gauss and GL points for each. I personally find the term "interfaces" a bit confusing, because many of these are points in the interior of an element, and I would otherwise assume that interfaces are on the boundary between elements. I appreciate where this comes from, but wonder if you might find a less confusing term.

5) eqn (28) I would find it useful to explain here that this works due to the exact mapping from the CG to the DG space under the vertical derivative.

6) eqn (29) Please explain these boundary conditions and their relevance.

7) p18, line 10. Why no preconditioner for GMRES? Are there any implications in terms of scaling of iterations with resolution? Please report the number of iterations.

8) p24, bottom of page. Please explain the issue about CP staggering and transport a bit more - I didn't follow it. In our FEM with CP-type staggering, this doesn't appear to be an issue.

9) p25, please explain this statement: "In general, it is not recommended to use hyper-diffusion with a higher order than the dynamical discretization (bottom left) since the impact of the hyper diffusion will be in the truncation order of the method."

10) p28. "High-order vertical discretizations are typically associated with strong oscillations that can induce perturbations that grow into unstable eddies." Please emphasise that this is only true when advection is present and discretised using a central/unstabilised scheme.

---

## Referee Comment (RC2) · D. Abdi (Referee) · 14 Feb 2016

This paper examines use of spectral elements in the vertical direction for the solution of the Non-Hydrostatic Euler equations using an Implicit-Explicit (IMEX) strategy. Staggering is important to avoid stationary modes, especially in the vertical direction which is often treated implicitly (e.g. 1D IMEX with columns). This paper nicely combines these two concepts to a method referred as Staggered Nodal Finite Element Method (SNFEM). The method is explained well and validated with standard benchmarks in atmospheric modeling. Therefore, I strongly recommend this work for publication.

Here are some comments and questions

1. Page 3, Line 20: "The fact that LGL points are clustered optimally near element edges lends itself to better resolution of the ABL...", is mentioned as an advantage of SNFEM. It seems to me that is only the case if one element is used in the vertical. If two or more elements are used, there will be clustering of nodes in the middle of the domain – which results in non-optimal placement of nodes overall. Moreover, sufficiently resolving the boundary layer probably requires many near-wall elements – elongated due to the anisotropic refinement that will in-turn degrade accuracy of solution.

2. Page 31, Line 20: I would have liked to see the baroclinic or similar test case done on the sphere (curved geometry). I understand this is postponed for a future work but I am curious to see if a high-order vertical discretization would be enough to maintain hydrostatic balance on the sphere without using reference states. This was mentioned as an advantage of going high order in Page 4, Line 20.

3. Page 30, Line 15: The paper recommends use of $4^{th}$ order polynomials in the vertical for reasons of efficiency and stabilization of the spectral element method. I wounder if $4^{th}$ order polynomial , which has 2 or 3 nodes near the boundary layer, be enough to resolve the ABL (see my first point).

4. Page 20, Line 15: Are there alternatives to staggering – some kind of interpolation scheme that would be equally efficient as staggering to avoid problems with stationary modes? I ask this because once we used discontinuous Galerkin methods in one of our projects in both horiztonal/vertical directions, and I believe all thermodynamic variables were collocated.
* * *

---

## Referee Comment (RC3) · Anonymous Referee #3 · 22 Feb 2016

The manuscript "A high-order staggered finite-element vertical discretization for non-hydrostatic atmospheric models" by J.E. Guerra and P. Ullrich describes the numerical formulation of a staggered nodal finite-element method for the vertical coordinate in nonhydrostatic models of atmospheric dynamics. The method is tested on benchmark cases of small and mesoscale flows. The paper is very well written and the results presented are comprehensive, the content is well structured and the analysis and discussion of the results are thorough and systematic.

However, given the validating character of the test cases presented, more could be done in order to assess the accuracy and efficiency properties of the method as well as its motivation and limitations. Once the comments listed below are addressed I am

[Figure]

happy to recommend publication of the manuscript in GMD.

**General comments**

1. The context where the authors place their study could be explored more in depth. Although only small to mesoscale test cases in Cartesian geometry are considered here, some scope could be provided to the reader on the eventual application of the present method. Do the authors target global or limited-area models, with adaptive grids or otherwise?

2. No reference is made in the manuscript to parallel developments of mixed finite element formulations, see e.g. Cotter and Shipton 2012, Cotter and Thuburn 2014, Thuburn and Cotter 2015. It would be useful to the reader if the authors clarified the points of contact of their manuscript with the mixed finite element literature, in particular regarding the different requirements in terms of artificial viscosity.

3. Some more formal accuracy validation could be useful in evaluating the performance of the method. Besides the data in Table 2 and the differences in Figure 9, convergence or self-convergence tests could be performed for example in the Straka or bubble case that will reproduce the theoretical accuracy of the method, see for example the graphs in Restelli and Giraldo 2009. It would be enough to perform a test at an earlier final time than the one used in the run (e.g., 300 s for Straka), ie before grid-scale effects kick in.

4. In terms of computational performance, it would be useful to see for all the test cases the results in Table 3. Is there a reason why the authors report results for a different number of cores? Aside of scalability not being optimal, the stress there appears to be on the effect of higher vertical order on performance. If the

emphasis is on testing scalability then the method should be tested on a much larger number of cores.

5. At least for the bubble runs, it would be helpful to test mass, momentum, and energy conservation properties of the method as seen for example in time series of these quantities throughout the simulation. What do the authors expect?

6. A previous study by the authors (Ullrich and Guerra 2015) is not referenced in the manuscript. Is the present manuscript an extension of that one? Some discussion on this point should be added in the introduction.

**Specific comments**

1. Page 3, lines 25 ff.: please list the test cases in the order they are presented in the paper.

2. Page 4, the objectives are probably best listed before the paragraph on the test cases.

3. Page 5, since all the test cases are in Cartesian coordinates, the authors should ask themselves whether the paper could gain conciseness and clarity by reducing or eliminating the discussion on arbitrary coordinate frames. At least the formulation in Cartesian coordinates could be reported at the end of Section 2 for the reader's convenience.

4. Notation, please use the 'g' symbol only once. It is currently used for the local coordinate vectors, the metric tensor, as well as for gravity in expression (13), the flux correction functions in expression (33), and for equation (48).

5. Section 3.2.2. is very detailed, especially in comparison with section 3.2.5., and the information contained in the latter appears somehow inconspicuous despite

its importance. Some material could be relegated to an appendix and Section 3.2. slightly restructured to better guide the reader from the aims of the paper to the tests.

6. Section 3.3, some more detail should be provided to the reader regarding CFL limitations of the employed time integration strategy. The information about the time splitting at page 31 should be placed in this section instead. Is the method second order in time? Again, a convergence test could be useful in backing up this claim.

7. Page 18, the list of choices is very clear, however point (5) is unnecessarily iterated in the tests, e.g. at page 21 line 25.

8. Page 21, lines 18 ff. Does the sponge layer cover 10 out of 25 km in the vertical, i.e. is 40% of the computational power just used for the sponge? Is this sponge layer the same as the Rayleigh layer discussed at page 23? If so please use only one of the two terms.

9. Section 4.3 could include standard 1d cuts at 5000 m height as seen in the literature as well as the convergence tests suggested above.

10. Point 1 page 31, was the analysis of wave dispersion not claimed to be left for future work on page 4?

11. Figure 3, please define 'VO' in the caption as well for clarity.

12. Figure 7 contains discussion of the results that should be put in the text instead.

13. Legibility of the lines in Figure 10 could be increased by either plotting the differences with respect to the reference or splitting the four lines into two panels.

14. Figures' captions should contain information on the contour interval values.

**Technical corrections**

- Expression (35) page 12: please replace $\partial g / \partial x (x \to 1)$ with $\lim_{x \to 1} \partial g / \partial x$.

- Page 18, line 11, 2 x finer $\to$ twice as fine as.

- Page 20, line 1, "in are" $\to$ 'are'.

- Page 25, what is a "truncation order"?

- Figure 1, should $v_{np}$ be $n_{vp}$ instead?

**References**

Cotter, C.J. and Shipton, J. Mixed finite elements for numerical weather prediction. Journal of Computational Physics 231, 7076–7091, 2012.

Cotter, C.J. and Thuburn, J. A finite element exterior calculus framework for the rotating shallow-water equations. Journal of Computational Physics 257, 1506-1526, 2014.

Restelli, M. and Giraldo, F.X. A conservative discontinuous Galerkin semi-implicit formulation for the Navier-Stokes equations in nonhydrostatic mesoscale modelling. SIAM Journal of Scientific Computing 31, 2231–2257, 2009.

Thuburn, J. and Cotter, C.J. A primal–dual mimetic finite element scheme for the rotating shallow water equations on polygonal spherical meshes. Journal of Computational Physics 290, 274–297, 2015.

Ullrich, P.A. and Guerra, J.E. Exploring the effects of a high-order vertical coordinate in a non-hydrostatic global model. Procedia Computer Science 51, 2076–2085, 2015.

---

## Short Comment (SC1) · 25 Feb 2016

Thank you Colin for your feedback. I would like to comment/reply to your points here:

1) You're absolutely correct here. Another review pointed this out and we will make the omission. Indeed, the solution fields depend on the basis and not on the local resolution due to node placement within an element. We have added "grid stretching" to our code recently in order to implement grid clustering correctly. This feature is currently being tested and is NOT included in the results for this paper.

2) We do cover this point in the validation section in the 6th assumption stated. However, we may repeat this in the introduction to make things clearer.

[Figure]

3) We do touch on the trade-off when increasing vertical order in a straight forward way. (Pg. 30, line 3) The code is undergoing an overhaul that includes improvements to the data structures, enabling vectorization, and improving parallelization. This is currently being tested.

4) We agree that "interfaces" is not optimal. However, it does serve its purpose and maps well to the nomenclature chosen for the interpolation and differentiation operators. We could use the acronyms "GLL" or "LGL" to refer to these locations.

5) Agreed.

6) We apply boundary conditions to kinematic variables only. No-flux conditions are used at the top and bottom boundary respectively.

7) We are still testing/researching various partitioning strategies so we have not settled on any one set where we can work on developing an appropriate preconditioner. However, our code does implement a direct solver on the generalized and diagonalized Jacobian (computed analytically or numerically). We settled on the GMRES without a preconditioner for this study to be consistent for all test cases, and because we observe that it is the most reliable solver.

8) This point may appear subtle, but arises from the vertical momentum equation in advective form as given in the text. Using the product rule you can write the transport term equivalently eliminating the vertical derivative of potential temperature. We call that "theta-flux" form and that corrects the issue in the Charney-Phillips configuration. A variational treatment of the equations involving integrals would also not suffer from this. We felt it necessary to include this observation because it is an example of staggering effects on the nonlinear equations as opposed to what is known for linear/wave dynamics.

9) A simple answer: we don't want to take 4 derivatives of a 3rd order polynomial to compute the hyperviscosity operator.

10) Correct and we will add language here to specify that this is true for nonlinear problems with advection, no explicit dissipation, and centered schemes.

---

## Short Comment (SC2) · 25 Feb 2016

Thank you Daniel for your feedback. I would like to reply to your comments here:

1) and 3) We will be removing this from the paper. The fields depend on the order of the basis functions within the element and not on the placement of the nodes. However, we are adding a grid stretching feature to Tempest. This way, grid clustering (of elements) near the boundary can be specified directly.

2) We do have a variety of tests running on the sphere, including the baroclinic wave. We're in the process of verifying these before we go into a similar study into the effects of the high-order vertical coordinate. For larger scale and tests on the sphere we will

be looking closely at how closely balanced states are maintained. Definitely more to come soon.

4) I will look further into this point. Variables that are staggered in the grid allow the advection scheme to "see" $2\Delta z$ modes. I am not aware of an advection scheme that can accomplish this without the use of filters or dissipation targeted at the spurious modes. We'll discuss your question further.

---

## Short Comment (SC3) · 10 Mar 2016

Thank you for your thorough review of our work. I'd like to respond here to your general comments. The specific comments you gave and technical corrections will all be addressed upon revision of the paper.

About your general comments:

1) Tempest is intended and built to be a general framework for atmospheric simulation. This includes all relevant scales in both spherical and Cartesian domains with and without topography. We will include this context more explicitly into the language. Clarifying this point will help illuminate our use of the equations in arbitrary coordinates

as it benefits the implementation.

2) We clearly see this as an oversight and will make sure to incorporate a better literature review of mixed finite elements.

3) I will take the recommendation of using the bubble test at 300s to perform spatial convergence and temporal convergence tests. This is certainly an important piece of information that is needed to evaluate the quality of our method. We expect from experience with our simulations that these tests will be unremarkable and show self convergence near the theoretical rates.

4) The results we report in the validation section were done on 10+ to 100+ cores using a distributed node cluster available at UC Davis. However, this is a shared resource and node allocation is only partially under our control. So, it is quite difficult to have run-to-run consistency in the hardware we are allocated. That is the reason we used a small local machine to do a small time performance study. Another reason is that Tempest is not yet optimized for performance i.e. no preconditioner for the linear solver, and other code improvements currently underway to improve parallelization.

5) We will consider including mass and energy conservation results. We fully expect that mass conservation is achieved in our tests. Energy conservation requires an analysis of the interpolation/differentiation operators for staggered column configurations that have revealed some requirements in the implementation. Also, our explicit dissipation strategy is still subject to heuristic tuning of diffusion coefficients (mainly in the horizontal hyperdiffusion, but also for the vertical velocity dependent diffusion) so we expect that mild energy loss is present. Our goal is, of course, to find configurations that are stable with minimal diffusion.

6) We will include our previous study and description of the method. However, that work differs in the equation system used. The difference is in the use of covariant velocity components as prognostics. This has some important implications in the construction of the various operators and we have found the current system to be better suited to

the discretization.

---

## Short Comment (SC4) · 10 Mar 2016

Some preliminary results on the sphere are attached, including the Jablonowski Williamson baroclinic instability and the DCMIP2012 3D mountain-induced Rossby wave train. No reference state is used and the model is able to accurately retain hydrostatic balance.

Figure 1: Snapshots from the baroclinic wave test case at day 7 and 9 simulated on a c90 grid with $30$ vertical levels and $30$ kilometer model cap ($n_{vp} = 3$). The time step is chosen to be $\Delta t = 250$ s. Surface pressure is plotted in the upper row, 850 hPa temperature in the middle row and 850 hPa relative vorticity in the bottom row.

Figure 2: Snapshots from the mountain-induced Rossby-wave train wave at day 5 (top row), day 15 (middle row) and day 25 (bottom row) simulated on a $n_e = 30$ grid with $30$ vertical levels and $30$ kilometer model cap ($n_{vp} = 3$). Geopotential height and temperature at $700$ hPa are shown in the left and right column, respectively.

Figure 3: Snapshots from the mountain-induced Rossby-wave train wave at day 5 (top row), day 15 (middle row) and day 25 (bottom row) simulated on a c90 grid with $30$ vertical levels and $30$ kilometer model cap ($n_{vp} = 3$). Zonal and meridional wind at $700$ hPa are shown in the left and right column, respectively.

[Figure]

**Fig. 1.**

[Figure]

**Fig. 2.**

[Figure]

**Fig. 3.**

---

## Editor Comment (EC1) · S. Marras (Editor) · 14 Mar 2016

In the paper, and in their last reply to the second reviewer, the authors state that the code is able to preserve hydrostatic balance. However, they do not show any time evolution of the vertical velocity to demonstrate their statement. As the authors stress this point, a plot that shows the evolution of vertical velocity with time for a problem with zero initial velocity will be beneficial. An example is shown in the figure attached for a linear finite element solution.

[Figure]

[Figure]

**Fig. 1.** Hydrostatic equilibrium of an atmosphere at rest above steep topography. Left: computational grid. Right: filled contours of vertical velocity $w$ at $t_f = 56000$ s. Vertical velocity: $-1e - 12 \leqslant w \leqslant 1e - 11$ m s$^{-1}$.

**5. 2D numerical tests**

In the following sections, the FE-VMS algorithm presented in Section 3 is tested against a suite of six standard tests used in dynamical core development. We divide the runs in two subsets according to the physics of the problems. In Section 5.1, *Numerical Tests I*, we perturb the background atmosphere with thermal anomalies that vary in definition and size. These tests do not have analytic solution and the metrics that we use are based on comparison with the literature using symmetry considerations, front velocity of the moving thermal perturbation, and the magnitude of extrema. This set includes the rising thermal bubble in a large domain [52], the rising thermal bubble in a small domain [53], a modified density current of [54], and the density current of [55]. In Section 5.2, *Numerical Tests II*, we solve two mountain problems that have semi-analytic solution based on the linear theory of small perturbations [56].

**5.1. Numerical Tests I: thermally-induced flows**

Given that the analytical solution does not exist, it must be understood that these tests can only give a qualitative (and relative) information on the accuracy that one model can achieve in the simulation of dynamic events in a low Mach environment.

*Background state.* The background state is characterized by a neutral atmosphere with uniform potential temperature $\bar{\theta}$ and background pressure $\bar{p}$ in hydrostatic equilibrium satisfying Eq. (16) such that

$$\bar{p} = p_0 \left(1 - \frac{g}{c_p \theta_0} z\right)^{c_p/R}, \tag{22}$$

where the surface potential temperature and surface pressure are $\theta_0 = 300$ K and $p_0 = 10^5$ Pa. The equation of state (2) is used to derive $\bar{\rho}$:

$$\bar{\rho} = \frac{p_0^{R/c_p}}{R\theta_0} \bar{p}^{c_v/c_p}. \tag{23}$$

**5.1.1. Case 1: warm bubble in a large domain**

The convection of a warm bubble in a uniform environment has been widely used by different authors (e.g. [53,57,41,52]) to test their codes. Like [52] after [58], in *Case 1* a domain that extends within $[0, 20000] \times [0, 10000]$ m$^2$ is defined. A large bubble of radius $r_0 = 2000$ m and centered in $(x_c, z_c) = (10000, 2000)$ m is initially at rest and used to perturb the atmosphere at uniform $\bar{\theta} = \theta_0 = 300$ K. The perturbation is given as a linear function of $R = \sqrt{(x - x_c)^2 + (z - z_c)^2}$ by

$$\theta' = \begin{cases} 0, & \text{if } R > r_0, \\ A[1.0 - R/r_0], & \text{if } R \leqslant r_0, \end{cases} \tag{24}$$

where the oscillation constant is $A = 2$ K. The initial velocity field is zero everywhere. No-flux boundary conditions are set for all the boundaries.

*Results Case 1.* To compare directly against reference [52], the final time is set to $t_f = 1020$ s. We perform three runs on three different resolutions: (1) $\Delta x = \Delta z = 50$ m, (2) $\Delta x = \Delta z = 125$ m, and (3) $\Delta x = \Delta z = 250$ m. Fig. 2 shows the values of $\theta'$ and $p'$ for the two finest grids. For $\theta'$, the results qualitatively agree with those of [52], where pressure is not shown. However, quantitatively our results show a higher degree of diffusivity that can be quantified by the values in Table 2. A definitive construction of $\tau$ in VMS does not exist yet and a different definition could improve this results.

**Fig. 1.** vertical velocity

---

## Author Comment (AC1) · 22 Mar 2016

Dear Simone,

Here are the results you requested. These are 2 hour simulations using the Schär mountain profile with zero background wind. Here we are again looking at vertical orders 2, 4, and 10 in the same manner as the original manuscript. The fields are plotted relative to the initial state of the atmosphere, but the reference state is not being used in the computations. We observe that there is an improvement in the error with increasing vertical order, however that is limited by the constant 4th order horizontal terms. We note the presence of the Lorenz vertical mode which does vanish for Charney-Phillips.
* * *
[Figure]

**Fig. 1.** Vertical Order 2 with Lorenz

[Figure]

**Fig. 2.** Vertical Order 4 with Lorenz

[Figure]

**Fig. 3.** Vertical Order 10 with Lorenz

---

## Editor Comment (EC2) · S. Marras (Editor) · 24 Mar 2016

Dear Jorge,

Thank you for sending me those plots. The article would benefit from the inclusion of some of them. You often touch on the topic in the manuscript, but do not show results that can demonstrate your statement on this specific topic. It is not necessary that you add an entire discussion of it; one, as long as significant, plot would support the text.

Thank you.

Best, Simone Marras

---

## Author Response (AR1)

Manuscript prepared for Geosci. Model Dev. Discuss.
with version 4.1 of the LaTeX class copernicus_discussions.cls.
Date: 25 April 2016

**A High-order Staggered Finite-Element Vertical Discretization for Non-Hydrostatic Atmospheric Models: ANNOTATED REVISION TRACKING FILE AND REFEREE RESPONSES.**

**J. E. Guerra and P. A. Ullrich**

Jorge Guerra, Department of Land, Air and Water Resources, University of California, Davis, One Shields Ave., Davis, CA 95616. Email: jeguerra@ucdavis.edu

Correspondence to: Jorge E. Guerra
(jeguerra@ucdavis.edu)

Discussion Paper | Discussion Paper | Discussion Paper | Discussion Paper

**1 Responses to Reviewer Comments**

We would like to thank all reviewers again for their thorough reviews. We believe these comments have greatly improved the quality of the manuscript and opened up several avenues for future investigation. References in boldface red font throughoutt this difference manuscript point to comments from referees as they have been applied to specific sections. Here we present a collection of responses to these comments that reflect those made in the interactive discussion.

**1.1 Reviewer 1**

1. We agree on this point and removed this from the introduction.

2. A change was made in the text to emphasize that we use a constant number of levels.

3. We do touch on the trade-off when increasing vertical order in a straight forward way. (Pg. 30, line 3) The code is undergoing an overhaul that includes improvements to the data structures, enabling vectorization, and improving parallelization. We do see a substantial drop in the CFL number with higher order (see section 5). We are going to investigate this behavior while improvements to the time integration scheme are made.

4. We agree that "interfaces" is not optimal. However, it does serve its purpose and maps well to the nomenclature chosen for the interpolation and differentiation operators. We could use the acronyms "GLL" or "LGL" to refer to these locations, but believe that is equally cumbersome or worse.

5. We have made this change in the text.

6. We have made this change in the text.

7. We are still testing/researching various partitioning strategies so we have not settled on any one set where we can work on developing an appropriate preconditioner. However, our code does implement a direct solver on the generalized and diagonalized Jacobian

(computed analytically or numerically). We settled on the GMRES without a preconditioner for this study to be consistent for all test cases and allow for modification of the right-hand-side without having to update the analytical Jacobian calculation. Since this approach is very preliminary and not reflective of the final operational product, we did not
5   think it would be scientifically valuable to report on the number of iterations required in the GMRES solve.

8. See the updated discussion in the Straka Density Current section 5.3 where we have updated an explanation of this phenomenon. It is a consequence of employing the advective form of vertical $\theta$ transport when $\theta$ is stored in interfaces (Charney-Phillips).

10   9. We updated the text on this point. It pertains to taking more derivates (2 or 4 for diffusion and hyperdiffusion) than is supported by the underlying polynomial space used to reconstruct the flow fields.

**1.2 Reviewer 2**

1. Deleted this from the text. Similar issue from first reviewer.

15   2. We omit any test cases on the sphere for this work. However, these will be featured in a subsequent work very soon. Nonetheless we have added results from a still atmosphere over the Schär mountain in section 5.2 to show that we can approximate hydrostatic balance accurately.

3. Goes along with first point.

20   4. We will look further into this point. Even with staggering the advection scheme is unable to "see" $2\Delta z$ modes. We are not aware of a collocated advection scheme that can accomplish this without the use of filters or dissipation targeted at the spurious modes. In fact, as proven in Ullrich (2014), any scheme for solving the advection equation will have at least one non-trivial mode with zero phase velocity [Ullrich, P.A. (2014) "Understanding the
25   treatment of waves in atmospheric models, Part I: The shortest resolved waves of the 1D

Discussion Paper | Discussion Paper | Discussion Paper | Discussion Paper | Discussion Paper

linearized shallow water equations" Quart. J. Roy. Meteor. Soc., Volume 140, Issue 682, pp. 1426–1440, doi: 10.1002/qj.2226.]

**1.3 Reviewer 3 - General Comments**

1. See updated introduction and concluding remarks.

2. A reference is now made to these works in the introduction.

3. Convergence tests are done for the Thermal Bubble for a short period of time in the new subsection 4.5, which discusses the numerical properties of the method.

4. We are indeed testing for any stresses or limitations with the use of high order elements in the vertical. The data in Table 4 is a preliminary result intended to keep the underlying hardware constant. Scalability at this point is secondary, but we are currently developing a more strict parallel implementation and will make a detailed study of performance on a large distributed system. We expect to have this data in a subsequent work covering test cases on the sphere.

5. We tested total system mass, energy, and momentum for the Robert Bubble and present our findings in the new section 4.5 along with discussion.

6. The introduction has been updated with this reference.

**1.4 Reviewer 3 - Specific Comments**

1. The order of the test cases is changed in the introduction.

2. Objectives are now given up front prior to outlining the test cases used.

3. We have updated the presentation of the equations in section 3 to include the metric tensor matrices. The equations, including the coordinate generalizations, are presented as they are implemented in our simulations. We also added the semi-discrete set for the 2D Cartesian case with topography in section 4.2.5.

4. We retain the geopotential $\Phi$ (eliminating the gravitational acceleration $g_c$ from the text) and changed the flux reconstruction polynomials from $g$ to $G$ so that only the metric tensors and coordinate vectors are named $g$ and $\mathbf{g}$ throughout the text.

5. Section 4.2.5 has been augmented to bring together the discrete operators into a system that can be computed in a manner consistent with our software implementation. We maintain the level of detail in the definition of the interpolation and derivative operators since that is the major contribution from this work.

6. CFL results for the bubble cases are now given in Fig. 20 and discussed in section 4.5

7. Deleted this from the list of assumptions.

8. Changed all "sponge" to "Rayleigh layer". We tuned the size of the domain, the depth of the Rayleigh layer, and apply the Rayleigh strength gradually to best simulate the outflow boundary condition and have the least effect on the interior flow. In order to avoid wave reflection, the depth of the Rayleigh layer must at least be as large as the largest stationary vertical wavelength.

9. We include 1D cuts for the Density Current under section 5.3, Figs. 14 and 15 at 1200 meters elevation.

10. We reworded this conclusion to more accurately reflect what we can infer from low-order centered difference schemes and what we observe for higher order discretizations. The linear analysis will be presented in a later work.

11. Changed Figure 3 per this comment.

12. Moved disccusion in the caption of Figure 7 to the text.

13. Figure 10 has been replotted (same data) to be clearer and show a greater portion of the domain under the influence of the Rayleigh layer.

14. Contour intervals added to all contour plots.

**1.5 Reviewer 3 - Technical Corrections**

All of these are changed as suggested in the revised text.

[revised manuscript text omitted]

---

## Referee Report (RR1)

**Review of the revised manuscript "A high-order staggered finite-element vertical discretization for non-hydrostatic atmospheric models" by J.E. Guerra and P. Ullrich.**

The authors have addressed the comments in a satisfactory way and I recommend the revised paper for publication. I would like to draw the attention on some points:

1. Figure 20: it might be worth adding some detail in the text or the caption about the reference solution used to compute the errors (I assume it would be the result of a high-resolution run). Moreover, some comments may help the reader in understanding the reason behind the low rate of convergence in the spatial test, this being a paper on high-order methods.

2. I appreciate that the paper mostly concerns space discretization issues. However, I still struggle to understand some aspects of the time discretization. Regarding the Courant numbers in Table 3, my comment was aimed at understanding the theoretical stability threshold associated to the time discretization method on the one hand, and the Courant numbers relative to the simulated test cases on the other. Only the data for the bubble case are made available to the reader. The thresholds in Table 3 appear quite restrictive if, as the caption appears to suggest, they refer to acoustic Courant numbers. It would be helpful to report the maximum Courant number for all the simulations.

3. Figure 21: the oscillations in the vertical momentum graph are attributed to acoustics, do I understand it correctly then that over the course of the simulation the model resolves acoustic waves? It would be helpful to report the Courant number in this case as well.

4. Notwithstanding the reservations over the scalability results in Table 4, there does not seem to be a reference to the Table in the text of the paper.

---

## Author Response (AR2)

Manuscript prepared for Geosci. Model Dev. Discuss.
with version 4.1 of the LATEX class copernicus_discussions.cls.
Date: 10 May 2016

**A High-order Staggered Finite-Element Vertical Discretization for Non-Hydrostatic Atmospheric Models: ANNOTATED REVISION TRACKING FILE AND REFEREE RESPONSES REV. 2.**

**J. E. Guerra and P. A. Ullrich**

Jorge Guerra, Department of Land, Air and Water Resources, University of California, Davis, One Shields Ave., Davis, CA 95616. Email: jeguerra@ucdavis.edu

Correspondence to: Jorge E. Guerra
(jeguerra@ucdavis.edu)

**1 Responses to Reviewer Comments**

We would like to thank all reviewers again for their input. This response is given to comments on the first revision by reviewer 3. This iteration has allowed us to correct some errors in the reporting of Courant numbers for the combined method presented in this paper and we are grateful for the opportunity to address these.

**1.1 Reviewer 3 - Revised Manuscript Comments**

1. Specific details on the resolution of the self convergence tests added to the caption of Figure 20. We note in the text (first paragraph of section 5.5) that convergence rates are limited by the perturbation function for bubble tests.

2. We updated section 5.5 and Table 3 to first correct the Courant numbers corresponding exactly to the explicit method outlined in section 4.3 and compare theoretical limits using spectral elements on scalar advection equations. Our resulting maximum timesteps approach the limiting values for explicit methods used within Strang.

3. Sound waves are not filtered or damped from the computations and are present throughout all simulations. The governing equations also explicitly include acoustic terms. Propagation of acoustic waves is controlled solely by the linearly implicit component of the time integration.

4. We include a reference to the very preliminary results of Table 4. A more thorough performance/scaling evaluation will be made on the appropriate hardware once data structures and algorithms are optimized.

[revised manuscript text omitted]